# Protein sequence design with a learned potential

Namrata Anand [1], Raphael Eguchi [2], Irimpan I. Mathews[3], Carla P. Perez [4], Alexander Derry [5], Russ B. Altman[1,6] & Po-Ssu Huang [1✉]

The task of protein sequence design is central to nearly all rational protein engineering problems, and enormous effort has gone into the development of energy functions to guide design. Here, we investigate the capability of a deep neural network model to automate design of sequences onto protein backbones, having learned directly from crystal structure data and without any human-specified priors. The model generalizes to native topologies not seen during training, producing experimentally stable designs. We evaluate the generalizability of our method to a *de novo* TIM-barrel scaffold. The model produces novel sequences, and high-resolution crystal structures of two designs show excellent agreement with in silico models. Our findings demonstrate the tractability of an entirely learned method for protein sequence design.

[1] Department of Bioengineering, Stanford University, Stanford, CA, USA. [2] Department of Biochemistry, Stanford University, Stanford, CA, USA. [3] Stanford Synchrotron Radiation Lightsource, Menlo Park, CA 94025, USA. [4] Biophysics Program, Stanford University, Stanford, CA, USA. [5] Biomedical Informatics Training Program, Stanford University, Stanford, CA, USA. [6] Departments of Genetics and Medicine, Stanford University, Stanford, CA, USA. ✉email: possu@stanford.edu

Computational protein design has emerged as a powerful tool for rational protein design, enabling significant achievements in the engineering of therapeutics[1–3], biosensors[4–6], enzymes[7,8], and more[9–11]. Key to such successes is robust sequence design methods that minimize the folded-state energy of a pre-specified backbone conformation, which can either be derived from existing structures or generated *de novo*. This difficult task[12] is often described as the inverse of protein folding—given a protein backbone, design a sequence that folds into that conformation. The functional design of enzymes, ligand binding sites, and interfaces all require fine-grained control over side-chain types and conformations. Current approaches for fixed-backbone design commonly involve specifying an energy function and sampling sequence space to find a minimum-energy configuration[13–15], and enormous effort has gone into the development of carefully modeled and parameterized energy functions to guide design, which continue to be iteratively refined[16,17].

With the emergence of deep learning systems and their ability to learn patterns from high-dimensional data, it is now possible to build models that learn complex functions of protein sequence and structure, including models for protein backbone generation[18–20] and protein structure prediction[21,22]; as a result, we were curious as to whether an entirely learned method could be used to design protein sequences on par with energy function methods. Recent experimentally validated efforts for machine learning-based sequence generation have focused on sequence representation learning without structural information, requiring fitting to data from experiments or from known protein families to produce functional designs[23,24]. We hypothesized that by training a model that conditions on local backbone structure and chemical environment, the network might learn residue-level patterns that allow it to generalize without fine-tuning to new backbones with topologies outside of the training distribution, opening up the possibility for generation of *de novo* designed sequences with novel structures and functions. Structure-based machine learning methods for design thus far have focused on mutation prediction[25–30], rotamer repacking of native sequences[31], or amino acid sequence design without modeling side-chain conformers[32–35], with some experimental validation including circular dichroism data[33] and fluorescence[26]. We sought to build a model that could generalize to unseen backbones with no homologous sequence data included in the training, as well as to validate that designs fold into target structures with designed side-chain conformations. As such, we explored a method in which the neural network not only designs the sequence but explicitly builds rotamers and evaluates full-atom structural models, an approach not reported to date.

Conventional energy functions used in sequence design calculations are often composed of pairwise terms that model interatomic interactions. Given the expressivity of deep neural networks, or their ability to approximate a rich class of functions, we predicted that a model conditioned on chemical context could learn higher-order (multi-body) interactions relevant for sequence design (e.g., hydrogen bonding networks). Furthermore, most energy functions are highly sensitive to specific atom placement, and as a result, designed sequences can be convergent for a given starting backbone conformation. For most native proteins, however, the existence of many structural homologs with low sequence identity suggests that there is a distribution of viable sequences that can adopt a target fold, but the discovery of these sequences given a fixed-backbone reference structure is difficult. We hypothesized that a learned model could operate as a soft potential that implicitly captures backbone flexibility, producing diverse sequences for a fixed protein backbone.

In this study, we explore an approach for sequence design guided only by a neural network that explicitly models side-chain conformers in a structure-based context (Fig. 1A), and we assess its generalization to unseen native topologies and to a *de novo* TIM-barrel protein backbone. The model produces novel sequences, and the high-resolution crystal structures of two designs show excellent agreement with in silico models.

## Results

**Design algorithm.** We are interested in sampling from the true distribution of $n$-length sequences of amino acids $Y \in \{1\ldots20\}^n$ conditioned on a fixed protein backbone. The backbone is fully specified by the positions of each residue's four $N - C_\alpha - C - O$ atoms and the C-terminal oxygen atom, whose positions are encoded as $X \in \mathcal{R}^{(4n+1)\times3}$; thus, the final conditional distribution we are interested in modeling is:

$$P(Y|X) = p(y_{i=1}, \ldots, y_n|X) \tag{1}$$

Due to the local nature of the physical interactions within proteins, we can expect that the likelihood of a given side-chain identity and conformation will be dictated by neighboring residues. Defining $\text{env}_i$ as the joint distribution over backbone atoms $X$ and neighboring residues $y_{NB(i)}$ at a given residue position $i$, the conditional side-chain distribution at position $i$ can be factorized sequentially as follows:

$$p(y_i|X, y_{NB(i)}) = p(y_i|\text{env}_i) = p(r_i|\text{env}_i) \prod_{j=1}^{4} p(\chi_i^j|\chi_i^{1:j-1}, r_i, \text{env}_i) \tag{2}$$

where $r_i \in \{1\ldots20\}$ is the amino-acid type at position $i$ and $\chi_i^1, \chi_i^2, \chi_i^3, \chi_i^4 \in [-180°, 180°]$ are the torsion angles for the side-chain.

We train a deep neural network conditional model $\theta$ to learn these conditional distributions from data. Conditioning on the local environment, the network autoregressively predicts distributions over residue types $p_\theta(r_i|\text{env}_i)$ and rotamer angles $p_\theta(\chi_i^1|\hat{\chi}_i^{1:j-1}, \hat{r}_i, \text{env}_i)$, conditioning on native residue type $\hat{r}_i$ and rotamer angles $\hat{\chi}_i^{1:j-1}$ (Fig. 1B). Our design algorithm involves iteratively sampling side-chains conditioned on their local chemical environments from these network-predicted distributions. We can approximate the joint probability of a sequence $P(Y|X)$ by the pseudo-log-likelihood (PLL)[36] of the sequence under the model

$$\text{PLL}(Y|X) = \sum_i \log p_\theta(y_i|\text{env}_i) \tag{3}$$

which we optimize in order to find high-likelihood sequences under the model. Over the course of design, the algorithm builds full-atom structural models that can be evaluated using established structure quality metrics.

We use a 3D convolutional neural network as our classifier $\theta$, training the model on X-ray crystal structures of CATH 4.2 S95 domains[37–39], with train and test set domains separated at the topology level. For the amino-acid type prediction task, our conditional model achieves a 57.3% test set accuracy, either outperforming[40] or matching[26,41,42] previously reported machine learning models for the same task. The predictions of the network correspond well with biochemically justified substitutability of the amino acids (Supplementary Fig. 1A–C); this learned substitutability is a necessary feature for design, as we expect proteins to be structurally robust to mutations at many residue positions. The residue type-conditioned rotamer prediction module by construction learns the backbone-dependent joint rotamer angle distribution $p(\chi_i^1, \chi_i^2, \chi_i^3, \chi_i^4|X, r_i)$. The network-learned independent $\chi$ distributions match empirical residue-specific rotamer distributions, which rotamer libraries typically seek to capture (Supplementary Fig. 2).

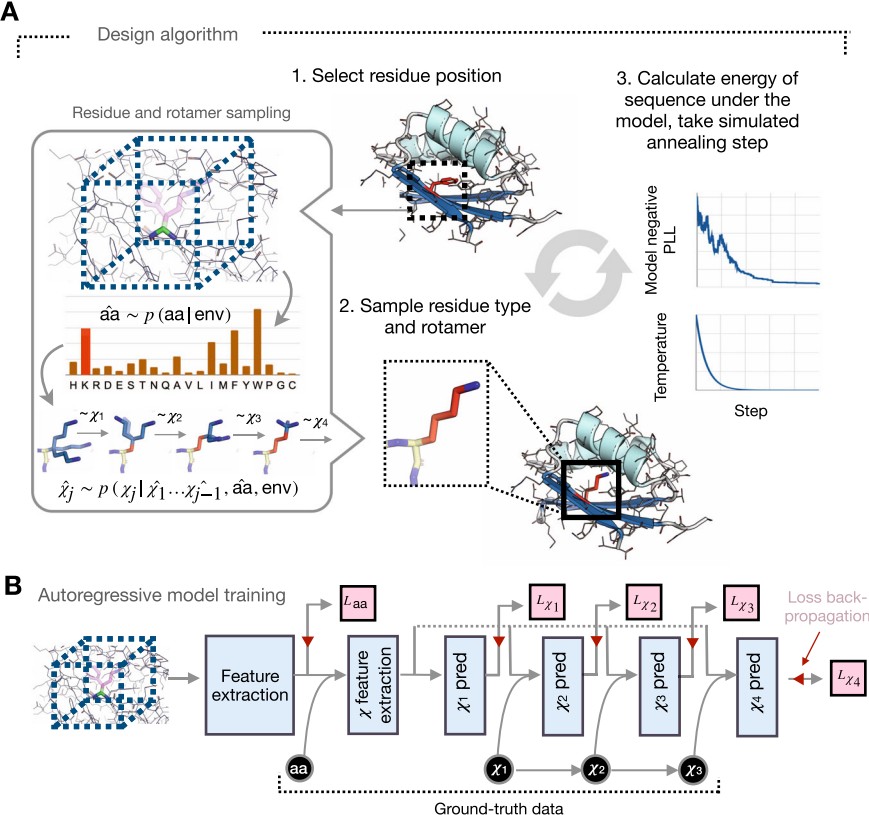

**Fig. 1 Fully learned sequence and rotamer design onto fixed protein backbones. A** Sequences are designed onto fixed protein backbones by (1) iteratively selecting a candidate residue position, (2) using a neural network model to sample amino-acid type and conformation, and (3) optimizing the negative pseudo-log-likelihood of the sequence under the model via simulated annealing. (Inset, left) Given the local chemical environment around a residue position (box, dashed, not to scale), residue type and rotamer angles are sampled from network-predicted distributions. **B** The neural network model is trained to predict residue identity and rotamer angles in an autoregressive fashion, conditioning on ground-truth data (black). The trained classifier predicts amino-acid type as well as rotamer angles conditioned on the amino-acid type. Cross-entropy loss objectives are shown in pink.

**Design algorithm generalizes to unseen backbone topologies.**
We sought to assess the degree of generalization of the algorithm to native backbones from the test set, which have CATH-defined topologies not seen by the model during training. We selected four test case backbones that span the major CATH classes—all alpha, alpha–beta, and all-beta (Fig. 2A). To validate the entirely learned approach, we challenged the model to fully redesign sequences given starting backbones. If the model has generalized, it should be able to recover native rotamers and sequence to a degree during design, as well as design key structural and biochemical elements typically seen in folded proteins.

Given the native sequences for the test cases, the model recovers native rotamers with high accuracy for the test case backbones across 5 design trajectories each (rotamer angles are within 20 degrees of the native angle on average 72.6% of the time) and with higher accuracy in the hydrophobic core regions of the protein (90.0% accurate within 20 degrees) (Fig. 2B, C and Supplementary Figs. 3A, B); this performance is on par with top benchmarked methods such as Rosetta[43] and another learned method for scoring rotamers[31]. Tasked with designing the sequence and rotamers from scratch (see "Methods" section), the model designs recapitulate between 25 and 45% of the native sequence in general, with a greater overlap in the buried core regions of the protein that are more constrained and therefore might accommodate a limited set of residue types (Fig. 2D and Supplementary Figs. 3C). The model designs are more variable in solvent-exposed regions, akin to sequences homologous to the native structure found through multiple sequence alignment (MSA) (Fig. 2E). Furthermore, the secondary structure prediction accuracy

for model-designed sequences are comparable to that of the native sequence (Fig. 2F and Supplementary Fig. 4C), indicating that despite the variation, the designed sequences retain local residue patterns that allow for accurate backbone secondary structure prediction.

Model designs tend to have well-packed cores (Supplementary Fig. 7A, B) and in general, the model-designed sequences tend not to have hydrophobic residues in solvent-exposed positions, likely due to the abundance of cytosolic protein structures available (Supplementary Fig. 7C). Additionally, the model designs match the native structure in terms of numbers of side-chain and backbone buried unsatisfied hydrogen bond acceptors and donors (Supplementary Fig. 7D–F); this indicates that over the course of model design, polar side-chains that are placed in buried regions are adequately supported by backbone hydrogen bonds or by the design of other side-chains that support the buried residue.

We also see a number of expected structural features across the test case designs, including placement of glycines at positive $\phi$ backbone positions (Supplementary Figs. 2G and 4D), N-terminal helical capping residues (Fig. 2H and Supplementary Figs. 4E), universal design of a proline at the cis-peptide position P21 for *1cc8* (Supplementary Fig. 4F), and, by inspection, polar networks supporting loops and anchoring secondary structure elements (Supplementary Fig. 5).

For each test case backbone, we selected 4 out of 50 designs for further characterization based on ranking by the model PLL and other metrics (Supplementary Tables 22–26). Importantly, this ranking uses no information about the native sequences. We validated these sequences by computational structure prediction

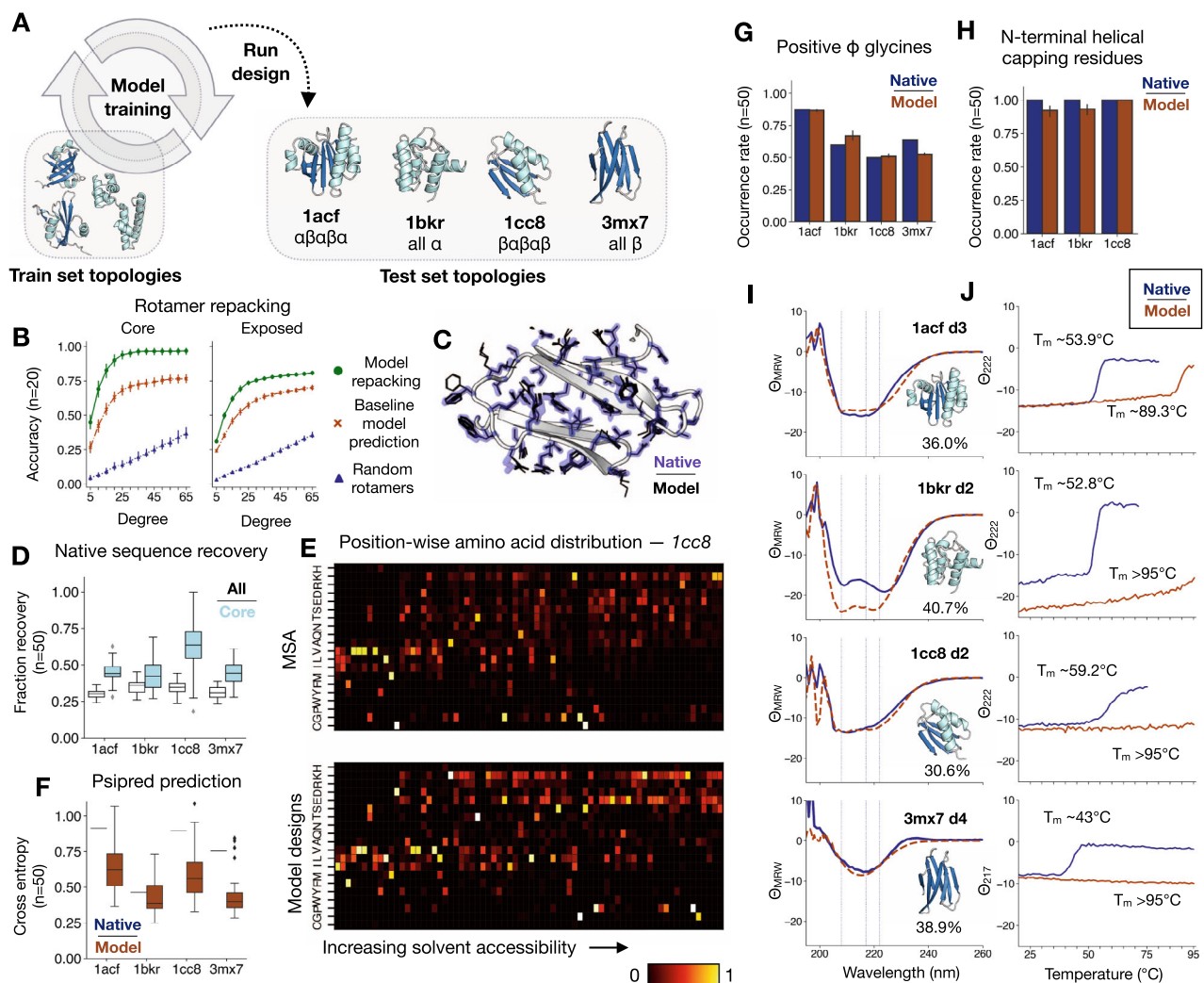

**Fig. 2 Generalization of model design to unseen topologies.** Data are presented as mean values ± 95% CI or as box plots with a median center, bounds of boxes corresponding to interquartile range (IQR), whisker length 1.5*IQR, and outliers rendered outside of this range. **A** The trained model is used to either repack rotamers or design entirely new sequences onto unseen test set structures with non-train-set CATH topologies. **B**, **C** Model-guided rotamer recovery for native test cases. **B** Rotamer repacking accuracy for buried core residues versus solvent-exposed residues as a function of degree cutoff. **C** 5 models superimposed with side chains shown as black lines compared to the native conformation shown in purple outline for test case *3mx7*. **D**–**H** Performance of sequence design onto test case backbones. **D** Native sequence recovery rate across 50 designs for all residues vs. buried core residues. **E** Position-wise amino-acid distributions for test case *1cc8*. Columns are normalized. (Top) Native sequence and aligned homologous sequences from MSA ($n = 670$). (Bottom) Model designs ($n = 50$). **F** Cross-entropy of Psipred secondary structure prediction from a sequence with respect to DSSP assignments[54,65–68,76]. **G** Fraction occurrence of glycines at positive $\phi$ backbone positions across test cases. **H** Fraction occurrence of N-terminal helical capping residues across designs for test cases with capping positions. **I**, **J** Far UV circular dichroism (CD) spectroscopy data for selected test case designs. **I** Mean residue ellipticity $\Theta_{MRW}$ ($10^3$ deg cm$^2$ dmol$^{-1}$) for CD wavelength scans at 20 °C for native structures (blue, dashed) vs. select model designs (orange, solid) *1acf d3*, *1bkr d2*, *1cc8 d2*, and *3mx7 d4*. Sequence identity to native reported within each panel. **J** Thermal melting curves for select model designs monitoring $\theta_{MRW}$ ($10^3$ deg cm$^2$ dmol$^{-1}$) at 222 nm or 217 nm for *3mx7*.

by the Rosetta ab initio application (Supplementary Figs. 9 and 10). The model designs achieved far better recovery than a 50% randomly perturbed control, suggesting that close recovery of the native backbone is due to features learned by the model and not simply due to sequence identity with the native. Interestingly, the model designs converge on some sequence features that are not seen in the native sequence, yet appear in homologous sequences (Supplementary Fig. 6).

Given the method's strong performance under these sequence quality metrics, we sought further confirmation that the model designs would express and be soluble and folded. Of the 16 designs tested, 15 expressed well in bacteria, and 10 appeared well-folded under circular dichroism (CD) wavelength scans (Supplementary Figs. 11 and 12). For each test case, at least 1 of 4 designs appeared

folded and had the expected secondary structure signature under CD. For example, the top design for *1bkr* has 208 nm and 222 nm alpha-helical signal as expected for a helical bundle, while all of the *3mx7* designs have clear 217 nm beta-strand signal, but no alpha signal, consistent with all-beta proteins (Fig. 2I). CD spectra for the top model designs match the native spectra well, and the designs were found to be more thermally stable than the native as well (Fig. 2J and Supplementary Fig. 13). Overall, these results indicate that the neural network model generalizes to topologies that are strictly unseen by the model during training.

**Model captures sequence–structure relationship.** Unlike analytical energy functions for macromolecular design, the model is

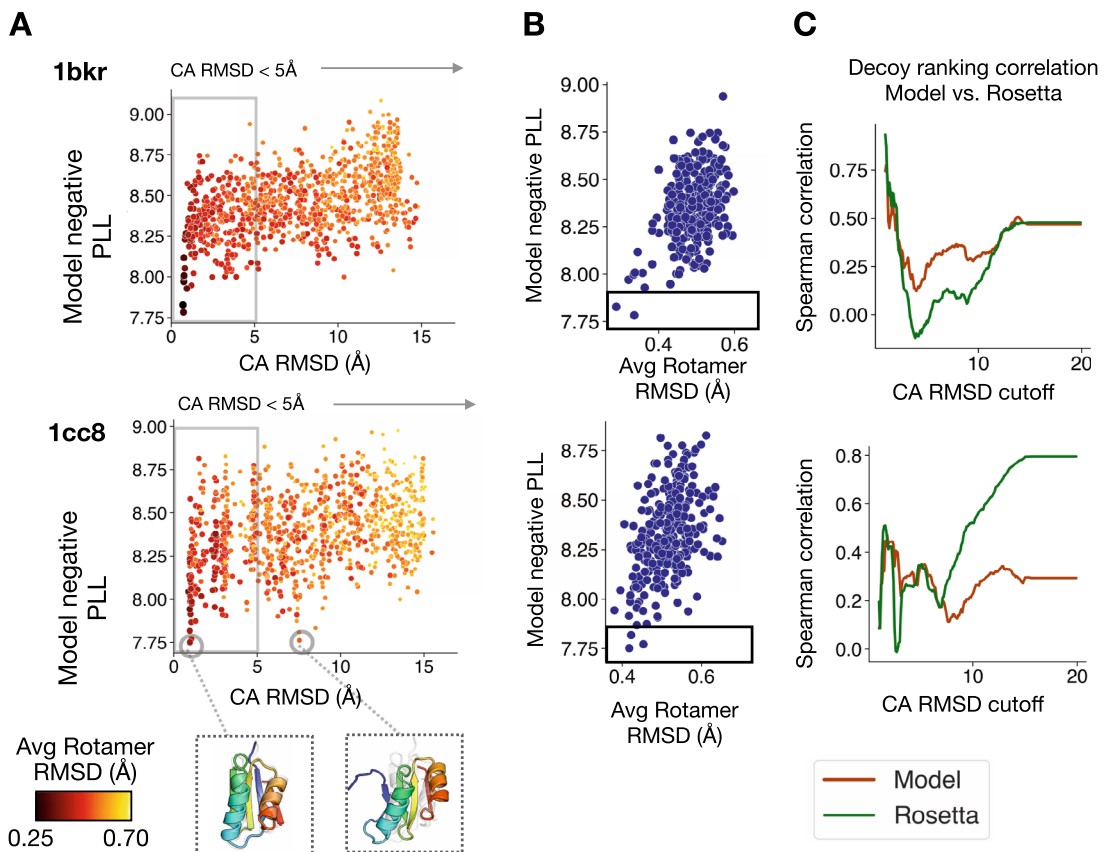

**Fig. 3 Model captures sequence–structure relationship. A**, **B** Decoy ranking by model negative pseudo-log-likelihood (PLL) of the native sequence. **A** Model negative PLL vs. alpha-carbon RMSD (Å) to the native structure for Rosetta ab initio decoys. Points are colored by average side-chain RMSD to native (Å). In some cases, the model assigns low negative PLL to high RMS backbones; for example, for *1cc8* an alternative pattern of beta-strand pairing is shown (Inset). **B** Model negative PLL of low backbone RMS structures (CA RMSD < 5 Å) vs. average side-chain RMSD (Å). Box highlights low model negative PLL assigned to low side-chain RMSD decoys. **C** Spearman rank correlation between model negative PLL or Rosetta energy vs. structure alpha-carbon RMSD (Å) as a function of increasing RMSD cutoff. In the low RMS regime (<5 Å), the model and Rosetta are able to rank low RMS structures to a similar extent.

not expected to generalize to structures far from the training distribution (e.g., unfolded, distended, or highly perturbed backbones), as it is trained by conditioning on the correct context. However, the model can in fact detect out-of-distribution backbones for a given sequence. Across a set of structures (decoys) generated via fragment sampling during Rosetta ab initio prediction, the model negative PLL is lowest for decoys with low RMSD to the native backbone (Fig. 3A and Supplementary Fig. 14A). There are cases where the model assigns low negative PLL to high RMS backbones (Fig. 3A and Supplementary Fig. 14A, inset); for example, for *1acf* an alternative N-terminal helix conformation and for *1cc8* an alternative pattern of beta-strand pairing are assigned low negative PLLs.

Additionally, in the low backbone RMS regime (<5 Å from the native), the model can further identify decoys with low RMSD side-chain conformations (Fig. 3B and Supplementary Fig. 14B). In essence, though the model is trained to learn sequence dependence on structure, it makes correct predictions about structures given sequence. Notably, the model demonstrates sensitivity to large perturbations, while remaining robust to small perturbations in the backbone (<2.5 Å) and side-chains (<0.40 Å), a desirable property for the design method. In the low RMS regime (<5 Å), the model is able to identify and rank low RMS structures to a similar extent as Rosetta (Fig. 3C).

**Model-based sequence design of a *de novo* TIM-barrel.** To assess whether the model could perform sequence design for *de novo* structures, we tested our method on a Rosetta-generated four-fold symmetric *de novo* TIM-barrel backbone, a $(\beta\alpha\beta\alpha)_4$ topology consisting of an eight-stranded barrel surrounded by eight alpha helices[44]. The design of *de novo* proteins remains a challenging task as it necessitates generalization to non-native backbones that lie near, but outside the distribution of known structures. Successful *de novo* protein designs often lack homology to any known native sequences despite the fact that *de novo* structures can qualitatively resemble known folds[44–46]. For a design protocol to perform well on *de novo* backbones it must therefore supersede simple recapitulation of homologous sequences. The TIM-barrel design case is of particular interest, as about ~10% of known enzymes are thought to adopt a TIM-barrel fold[47], making the fold a prime candidate for the rational design of enzymes and more generally for substrate binding.

We had the model fully redesign 50 four-fold symmetric sequences for the backbone and selected 8 designs for further characterization based on ranking by the model PLL and other metrics (see "Methods" section), using no information about previously confirmed sequences (Supplementary Table 27 and Supplementary Fig. 28). A subset of these 8 structures was predicted to fold with low backbone deviation (<4 Å) into the

target backbone by Rosetta (F2C, F4C) and trRosetta (F2C, F4C, F8C) (Supplementary Fig. 15).

Of these 8 designs, 4 are cooperatively folded proteins, as indicated by circular dichroism (CD) wavelength scans and by the presence of clear two-state transitions in thermal melt curves (Supplementary Figs. 16 and 17). All of the folded proteins designed by our model have higher thermal stability than the initial designs reported in the original study, with melting temperatures for 3 of the 4 ranging between 84 and 87 °C while the fourth protein (F5C) remains folded at 95 °C (Supplementary Fig. 17B). Folded sequences reported in the original study also require manual intervention to introduce a key aspartate residue required for the structure to cooperatively fold; while automated Rosetta design is unable to produce foldable sequences, our model produced the sequences with no manual intervention.

We successfully crystallized 2 model designs, F2C (1.46 Å resolution) and F15C (1.9 Å resolution), and the crystal structures validate that the sequences indeed fold into the TIM-barrel structure in agreement with the designed backbone (Fig. 4A and Supplementary Figs. 18A, 19A, Supplementary Table 31). For F2C, the C-terminal helix is dislodged, although the complete barrel is formed. We hypothesize that the design of Ala3 (and symmetric positions Ala49, Ala95, Ala141) reduces the hydrophobic volume in the core, possibly allowing other interactions in the crystallization conditions to dislodge the helix (further discussion in Supplementary Note 1). In the structure, we see that the C-terminal His tag for F2C interacts with the surface of the beta-barrel, which might also contribute to the dislodging of the C-terminal helix (Supplementary

Figs. 18A, E), although 3 of the 4 symmetric subunits are folded as expected. To elucidate this, we crystallized the protein with an N-terminal His-TEV tag (F2N, 1.58 Å resolution). Since the structure is a closed toroid, the N- and C-termini are proximal, and we see that the N-terminal tag is partially resolved folded against the barrel, displacing the C-terminal helix (Supplementary Fig. 18C). F15C adopts the full TIM-barrel fold, but for two of four monomers in the asymmetric unit of F15C, one of the β-α loops shifts to interact with an adjacent monomer in the crystal (Supplementary Fig. 19B, C). However, we note that F15C primarily elutes as a monomer in size exclusion chromatography (Supplementary Fig. 16), and the interactions observed in the crystal structure resemble crystal packing, rather than a stable dimer interface. Apart from the described deviations, the structures fold as expected.

The model designs tend to recapitulate previous core sequences for the region of the protein between the helices and outer part of the barrel (Supplementary Fig. 20A), as well as for the inner part of the barrel, although some model designs do introduce novel features into these regions (Supplementary Fig. 20B).

These similarities aside, the model designs are in general more varied than other previously characterized designs, predominantly due to variation in non-buried residue positions (Fig. 4B and Supplementary Fig. 20C, Supplementary Table 29). Of particular interest are mutations in the interfaces between the helices surrounding the barrel where, remarkably, the design algorithm is able to find novel structural features that have not been discovered by Rosetta and human-guided design methodologies[44,48] (Fig. 4C, D and Supplementary Fig. 21A, B).

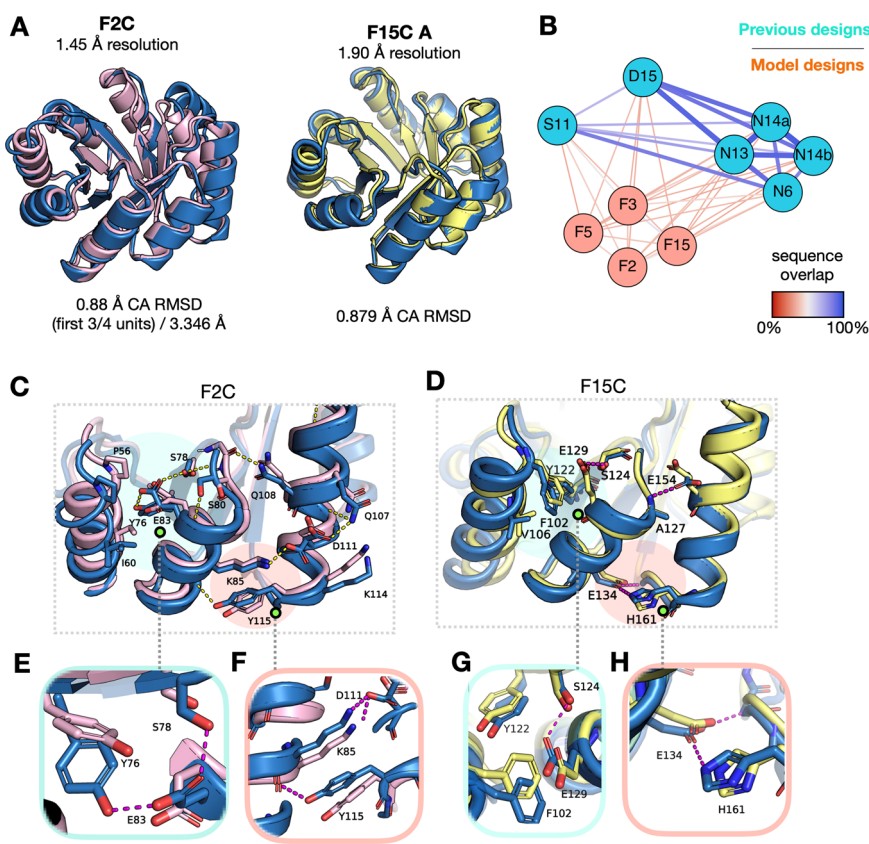

**Fig. 4 Model discovery of novel sequence features. A** Overlay of crystal structures (blue) with template TIM-barrel backbone for F2C (pink) and F15C (yellow). Alpha-carbon RMSD (Å) and sequence identity to sTIM-11 sequence[44] are given below structures. **B** Percent sequence identity (indicated by graph edge color and thickness) between TIM-barrel subunits for model TIM-barrel designs (orange) and previously characterized sequences for the same scaffold (blue), including sTIM-11 (*5bvl*, S11)[44], DeNovoTIM15 (*6wvs*, D15)[77], and DeNovoTIMs (N6, N13, N14a, N14b)[48]. N14a and N14b are two-quarters of the two-fold symmetric DeNovoTIM14. **C, D** Investigation of sequence features for the symmetric subunit near the top of the barrel (cyan shadow) and the helix interface between symmetric subunits (orange shadow) for **C** F2C and **D** F15C. Crystal structures are shown in blue overlaid with the design template (pink—F2C, yellow—F15C). **E–H** Closer inspection of novel sequence features designed by the model.

For F2C, the model uses an isoleucine to wedge between the long and short helices and forming a Tyr-Glu-Ser polar network that extends to coordinate the loop from the beta-sheet to the short helix (Fig. 4E); for F15C, the model has a different solution, designing phenylalanine and valine residues (F10, V14) to pack the long helix against the shorter one near the top of the barrel (Fig. 4G). Particularly interesting are the Tyr-backbone contact and the His-Glu-backbone polar network for F2C and F15C, respectively, that help coordinate the helices across symmetric subunits at the base of the structure; these are the only hydrogen bonding solutions designed at this position among all previous structure-confirmed sequences for this scaffold (Fig. 4F, H).

## Discussion

Our results demonstrate that a design algorithm guided by an entirely learned neural network potential can generate viable sequences for a fixed-backbone structure, generalizing to unseen topologies and *de novo* backbones. The method is flexible: the design protocol easily allows for adding position-specific constraints during design, and other neural network models such as graph networks or rotationally equivariant networks, could be used in place of the classifier network presented without fundamentally changing the method. Though presented in a stand-alone manner, in practice this method could be used in tandem with energy function-based methods for design, for example by using the model as a proposal distribution, while optimizing an external energy function. We anticipate that this type of approach could be used for the design of interfaces, protein–nucleic acid complexes, and ligand binding sites.

Notably, the design algorithm reflects key characteristics of energy functions, such as the ability to (1) accurately determine side-chain conformations, (2) differentiate the hydrophobic interior and polar exterior of the proteins, and (3) design hydrogen-bonding networks. With classical molecular mechanics force-fields, capturing these effects require terms that accurately describe Van der Waals, solvation, hydrogen bonding, as well as many other interactions; it also likely requires an independent hydrogen-bonding network search algorithm[49] and discrete side-chain representations from rotamer libraries. None of these are required with this approach. In contrast to energy function development, the model took only hours to train.

In this study, we sought to tackle key challenges with machine learning-based protein sequence design, including generalization to new folds and angstrom-level recovery of target structures. The learned neural network potential is able to guide high-dimensional sampling and optimization of the structure-conditioned sequence distribution. Our results show the practical applicability of an entirely learned method for protein design, and we believe this study demonstrates the potential for machine learning methods to transform current methods in structure-based protein design.

## Methods

**Model training**. Our model is a fully convolutional neural network, with 3D convolution layers followed by batch normalization[50] and LeakyReLU activation. We regularize with dropout layers with dropout probability of 10% and with L2 regularization with a weight $5 \times 10^{-6}$. We train our model using the PyTorch framework, with orthogonal weight initialization[51]. We train with a batch size of 2048 parallelized synchronously across eight NVIDIA v100 GPUs. The momentum of our BatchNorm exponential moving average calculation is set to 0.99. We train the model using the Adam optimizer ($\beta_1 = 0.5$, $\beta_2 = 0.999$) with learning rate $7.5 \times 10^{-5}$[52]. We use the same model architecture and optimization parameters for both the baseline (prediction from backbone alone) and conditional models (Supplementary Data 1).

Our final conditional classifier is an ensemble of four models corresponding to four concurrent checkpoints. Predictions are made by averaging the logits (unnormalized outputs) from each of the four networks. Trained models are

available at https://drive.google.com/file/d/1cHoyeI0H_Jo9bqgFH4z0dfx2s9as9Jp1/view?usp=sharing.

**Data**. To train our classifier, we used X-ray crystal structure data from the Protein Data Bank (PDB)[37], specifically training on CATH 4.2 S95 domains[38,39]. We first applied a resolution cutoff of 3.0 Å and eliminated NMR structures from the dataset. We then separated domains into train and test sets based on CATH topology classes, splitting classes into ~95% and 5%, respectively (1374 and 78 classes, 53,414 and 4372 domains each, see Supplementary Data 1). This ensured that sequence and structural redundancy between the data sets was largely eliminated. During training, we did not excise domains from their respective chains but instead retained the complete context around a domain. When a biological assembly was listed for a structure, we trained on the first provided assembly. This was so that we trained primarily on what are believed to be functional forms of the protein macromolecules, including in some cases hydrophobic protein–protein interfaces that would otherwise appear solvent-exposed.

The input data to our classifier is a $20 \times 20 \times 20$ Å$^3$ box centered on the target residue, and the environment around the residue is discretized into voxels of volume 1 Å$^3$. We keep all backbone atoms, including the $C_\alpha$ atom of the target residue, and eliminate the $C_\beta$ atom of the target residue along with all of its other side-chain atoms. We center the box at an approximate $C_\beta$ position rather than the true $C_\beta$ position, based on the average offset between the $C_\alpha$ and $C_\beta$ positions across the training data. For ease of data loading, we only render the closest 400 atoms to the center of the box.

We omit all hydrogen atoms and water molecules, as well as an array of small molecules and ions that are common in crystal structures and/or possible artifacts (Supplementary Data 1). We train on nitrogen (N), carbon (C), oxygen (O), sulfur (S), and phosphorus (P) atoms only. Ligands are included, except those that contain atoms other than N, C, O, S, and P. Bound DNA and RNA are also included. Rarer selenomethionine residues are encoded as methionine residues. For the baseline model, we omit all side-chain atoms while training, so that the model conditions only on backbone atoms. For the conditional model, the input channels include: atom type (N, C, O, S, or P), indicator of backbone (1) or side-chain (0) atom, and one-hot encoded residue type (masked for backbone atoms for the center residue). For the baseline model, the input channels only encode atom type, since all atoms are backbone atoms and we assume no side-chain information is known.

We canonicalize each input residue environment in order to maximize invariance to rotation and translation of the atomic coordinates. For each target residue, we align the N-terminal backbone $N - C_\alpha$ bond to the $x$ axis. We then rotate the structure so that the normal of the $N - C_\alpha - C$ plane points in the direction of the positive $z$ axis. Finally, we center the structure at the effective $C_\beta$ position. By using this strategy, we not only orient the side-chain atoms relative to the backbone in a consistent manner (in the positive $z$ direction), but also fix the rotation about the $z$ axis. We then discretize each input environment and one-hot encode the input by atom type.

**Design algorithm set-up**. We would like to sample from the probability distribution of $n$-length sequences of amino acids $Y \in \{1 \ldots 20\}^n$ conditioned on a fixed protein backbone configuration. The backbone is specified by the positions of the residues' non-hydrogen atoms whose positions are encoded as $X \in \mathcal{R}^{(4n+1) \times 3}$; thus, the final conditional distribution we are interested in modeling is $P(Y|X) = p(y_{i=1}, \ldots, y_n|X)$.

This is a high-dimensional structured probability distribution, and we make the following simplifying assumption to make the task of learning this distribution from data more tractable: identity and conformation (rotamer) of each side-chain $y_i$ is independent of all other side chains $y_j$ conditioned on the identity of the side chains in its neighborhood $y_{NB(i)}$.

This assumption motivates the use of a Conditional Markov Random Field (MRF) to model the target distribution, wherein the nodes of the MRF correspond to the residues and rotamer configurations, edges between nodes indicate the possibility of correlation between the residues or rotamers, and each node is conditionally independent of all other nodes, conditioned on the nodes in its Markov Blanket. More precisely, the input backbone $X$ defines a graph structure, where nodes correspond to side chain $y_i$ and edges exist between pairs of residues $(y_i, y_j)$ if and only if the corresponding backbone atoms are within some threshold distance of each other. Defining env$_i$ as the joint distribution over backbone atoms $X$ and the spatial neighborhood $y_{NB(i)}$, we see that the conditional distribution of residue and rotamers at a single position $i$ can be factorized sequentially as follows:

$$p(y_i|X, y_{NB(i)}) = p(y_i|\text{env}_i) = p(r_i|\text{env}_i) \prod_{j=1}^{4} p(\chi_i^j|\chi_i^{1:j-1}, r_i, \text{env}_i) \quad (4)$$

where $r_i \in \{1 \ldots 20\}$ is the amino-acid type of residue $i$ and $\chi_i^1, \chi_i^2, \chi_i^3, \chi_i^4 \in [-180°, 180°]$ are the torsion angles for the side-chain.

We train a deep neural network conditional model $\theta$ in an autoregressive manner to learn these conditional distributions from data. We also train a baseline model that has only backbone atoms as an input, as a means to initialize the backbone with a starting sequence.

**Sampling procedure**. Given a residue position $i$ and a local environment $env_i$ around that residue with either just backbone atoms (baseline model) or other residue side-chains as well (conditional model), the sampling procedure is as follows. First, sample a residue type $\hat{r}_i \in \{1 \ldots 20\}$ conditioned on the environment. Then, conditioned on the environment and the sampled amino-acid type, sample rotamer angle bins $\hat{\chi}_i^j \in [-180°, 180°]$ for $j = 1 \ldots 4$ in an autoregressive manner. The model in this instance learns a distribution over 24 rotamer bins (7.5° per bin). After a discrete rotamer bin has been sampled, the final rotamer angle $\tilde{\chi}_i$ is sampled from a uniform distribution over the bin.

$$\hat{\chi}_i^j \sim p_\theta(\chi_i^j | \hat{\chi}_i^{1:j-1}, \hat{r}_i, env_i) \tag{5}$$

$$\tilde{\chi}_i^j \sim \text{Unif} \left[ \text{BinLeft}(\hat{\chi}_i^j), \text{BinRight}(\hat{\chi}_i^j) \right] \tag{6}$$

Note that the subsequent autoregressive step conditions on the discrete rotamer bin, not the sampled continuous rotamer angle.

As residues and rotamers are sampled at different positions along the protein backbone, we monitor the negative pseudo-log-likelihood (PLL) of the sequence

$$\text{PLL}(Y|X) = \sum_i \log p_\theta(y_i | env_i) = \sum_i \log p(r_i | env_i) + \sum_{j=1}^4 \log p(\chi_i^j | \chi_i^{1:j-1}, r_i, env_i) \tag{7}$$

as a heuristic model energy. Note that most residues have fewer than four $\chi$ angles. At sampling time, after a residue type has been sampled, only the corresponding $\chi$ angles for that residue are sampled. For residues that do not have a particular $\chi^j$, an average value for the log probability of $\chi^j$ under the conditional model across the train set and across the rotamer bins is used instead in the PLL calculation.

In this study, we do simulated annealing to optimize the average PLL across residue positions. The acceptance criterion for a step is

$$p_{accept} = \min \left( 1, \frac{\exp\left(-\frac{1}{N}\sum_{i=1}^N \log p_\theta(y_i | env_i)\right)}{T} \right) \tag{8}$$

where temperature $T$ is annealed over the course of a design run.

In order to speed up convergence, we do blocked sampling of residues. In practice, we draw edges in the graph between nodes where corresponding residues have $C_\beta$ atoms that are less than 20 Å apart, guaranteeing that non-neighboring nodes correspond to residues that do not appear in each other's local environments. During sampling, we use greedy graph coloring to generate blocks of independent residues. We then sample over all residues in a block in parallel, repeating the graph coloring every several iterations. Additionally, we restrict the model from designing glycines at non-loop positions, based on DSSP assignment[53,54].

**Implementation and runtime**. The algorithm is implemented in Python, using PyTorch for loading and evaluating the trained networks and PyRosetta for building structural models based on the network predictions for residue type and rotamer angles. Code for the algorithm is available at https://github.com/ProteinDesignLab/protein_seq_des.

The runtime for our method for sequence design is determined primarily by two steps: (1) sampling residues and rotamer angles and (2) computing the model energy (negative PLL). These times are determined by the speed of the forward pass of the neural network, which is a function of the batch size, the network architecture, the GPU itself, and the number of GPUs used in parallel. Note that to compute the model energy, a forward pass of the network is done at each residue position about which the environment has changed.

Annealing for 2500 steps takes between 1 and 3 h for the native test cases on a computer with 32 GB RAM and on a single GeForce GTX TITAN X GPU, with up to 3 design runs running on the same machine/GPU in parallel (Supplementary Data 2). *Rosetta-RelaxBB* takes 20–30 min per design, while *Rosetta-FixBB* takes 5–15 minutes per design. Compressing the network or modifying the architecture and parallelizing the sampling procedure across more GPUs would improve the overall runtime. In addition, a faster annealing schedule and early stopping of optimization would also reduce the runtime.

**Native test case rotamer repacking experiments**. Five rounds of rotamer repacking were done on each of the four test case backbones (Supplementary Movie 1). Repacking is done by fixing the native sequence and randomizing starting rotamers, or using baseline model predictions on backbone atoms only to initialize rotamers. Rotamer prediction at each step and each residue position conditions on the true native residue identity. Model negative PLL averaged by protein length was annealed for 2500 iterations with starting temperature 1 and annealing multiplicative factor 0.995. Commands to reproduce experiments are provided in Supplementary Data 2.

**Native test case sequence design experiments**. Native test case structures *1acf* (2.00 Å), *1bkr* (1.10 Å), *1cc8* (1.022 Å), and *3mx7* (1.76 Å) belong to the beta-lactamase (CATH:3.30.450), T-fimbrin (CATH:1.10.418), alpha–beta plait

(CATH:3.30.70), and lipocalin (CATH:2.40.128) topology classes, respectively. We selected these test structures because they span the three major CATH[38,39] protein structure classes (mostly alpha, alpha–beta, and mostly beta) and because their native sequences were recoverable via structure prediction with Rosetta ab initio, ensuring they could serve as a positive control for later in silico folding experiments. Fifty rounds of sequence and rotamer design were done on each of the four test case backbones (Supplementary Movie 2). Sequences were initialized via prediction by the baseline model given backbone atoms alone. Model negative PLL averaged by protein length was annealed for 2500 iterations with starting temperature 1 and annealing multiplicative factor 0.995. Commands to reproduce experiments are provided in Supplementary Data 2. Sequences for further characterization were first filtered by specific criteria including all helices capped at the N-terminal, packstat ≥ 0.55 pre and post-RosettaRelax, and for some cases other cutoffs for side-chain and backbone buried unsatisfied hydrogen bond donors or acceptors (Supplementary Table 22). After filtering, sequences were ranked by model negative PLL for selection. No changes were made to the sequences produced by the model. Top sequences highlighted in Fig. 2I, J are *1acf d3*, *1bkr d2*, *1cc8 d2*, and *3mx7 d4*.

**TIM-barrel design experiments**. The TIM-barrel template backbone is a circularly permuted variant of the reported design sTIM-11 (PDB ID *5bvl*)[44], and the template was prepared using RosettaRemodel with sTIM-11 sequence, circularly permuted and without the originally designed cysteines which did not successfully form a disulfide (C8Q, C181V). Native TIM-barrels are included in the training set; however, *5bvl* is excluded from the training set. *5bvl* is distant in sequence and structure from any known protein, and contains local structural features that differ significantly from naturally occurring TIM-barrels[44]. The original design sTIM-11 was designed using a combination of Rosetta protocols and manual specification of residues. Fifty rounds of sequence and rotamer design were done on the TIM-barrel template backbone. Sequences were initialized via prediction by the baseline model given backbone atoms alone. Model negative PLL averaged by protein length was annealed for 2500 iterations with starting temperature 1 and annealing multiplicative factor 0.995. Residue and rotamer four-fold symmetry was enforced by averaging predicted logits across symmetric positions before normalizing to a discrete distribution and sampling. Commands to reproduce experiments are provided in Supplementary Data 3. Sequences for further characterization were filtered by the following criteria: all helices capped at the N-terminal and packstat ≥ 0.55 pre and post-RosettaRelax. 13 sequences were selected based on structure and sequence metrics (Supplementary Table 27) and from these 8 sequences were selected based on structural features and a set of simple criteria (Supplementary Table 28). No changes were made to the sequences produced by the model, except a single cysteine to valine mutation for F1 and F15 in each symmetric subunit (C5V) made ahead of testing for ease of purification, and the valine mutation was ranked highly by the model.

**Design baselines**. Rosetta design[55,56] is a method for sequence design that has been broadly experimentally validated.[55,56] We use Rosetta to design sequences in order to have a point of comparison with the model across the metrics used to evaluate design quality. Further observations about Rosetta performance can be found in Supplementary Note 1. The *Rosetta-FixBB* baseline uses the Rosetta packer[55], invoked via the *RosettaRemodel*[57] application, to perform sequence design on fixed backbones. This design protocol performs Monte Carlo optimization of the Rosetta energy function over the space of amino-acid types and rotamers[55]. Between each design round, side-chains are repacked, while backbone torsions are kept fixed. Importantly, the Rosetta design protocol samples uniformly over residue identities and rotamers, while our method instead samples from a learned conditional distribution over residue identities. The *Rosetta-RelaxBB* protocol is highly similar to the *Rosetta-FixBB* protocol but performs energy minimization of the template backbone in addition to repacking between design cycles, allowing the template backbone to move. Starting templates for both baselines have all residues mutated to alanine, which helps eliminate early rejection of sampled residues due to clashes. The REF2015 Rosetta energy function was used for all experiments[16,58]. In order to remove potentially confounding artifacts that emerge during construction and optimization of PDB structures, we additionally relax the test case backbones with constraints to the original atom positions under the Rosetta energy function and run design on these constrained relaxed backbones, as well as for the crystal structure. This step is necessary in order for the Rosetta protocols to in theory be able to recover the native sequence via optimization of the Rosetta energy function.

**Metrics**. To assess biochemical properties of interest for the designed sequences, we use the following three metrics: (1) packstat, a non-deterministic measure of tight core residue packing[59], (2) exposed hydrophobics, which calculates the solvent-accessible surface area (SASA) of hydrophobic residues[60], and (3) counts of buried unsatisfied backbone (BB) and side-chain (SC) atoms, which are the number of hydrogen bond donor and acceptor atoms on the backbone and side-chains, respectively, that are not supported by a hydrogen bond. We use PyRosetta implementations of these metrics. Backbone relaxes for designs were done with the

RosettaRelax application[61–64], with the ex1 and ex2 options for extra $\chi^1$ and $\chi^2$ rotamer sampling.

**Rosetta ab initio structure prediction**. Rosetta ab initio uses secondary structure probabilities obtained from Psipred[65–68] to generate a set of candidate backbone fragments at each amino-acid position in a protein. These fragments are sampled via the Metropolis–Hastings algorithm to construct realistic candidate structures (decoys) by minimizing Rosetta energy. Native test case runs used Psipred predictions from MSA features after UniRef90[69] database alignment. TIM-barrel design runs used Psipred predictions directly from sequence. All designs were selected without using any external heuristics, manual filtering, or manual reassignments. We obtained Psipred predictions, picked 200 fragments per residue position[70], and ran $10^4$ trajectories per design. Folding trajectories in Fig. 3, Supplementary Fig. 14 were seeded with native fragments.

**trRosetta structure prediction**. We also use the trRosetta online server to get structure predictions for designed sequences[22]. trRosetta uses a deep neural network to predict inter-residue distance and orientation distributions from sequence and multiple sequence alignment (MSA) data. These distributions are used to encode restraints to guide Rosetta structure modeling. No homologous structure templates were used for specifying distance constraints in the model-building step. We report alpha-carbon RMSD and GDTMM (Global Distance Test score using Mammoth for structure alignment) as another measure of structure correspondence.

**Protein purification**. Native test case proteins and FXN TIM designs were produced as fusions to an N-terminal 6xHis tag followed by a Tobacco Etch Virus (TEV) cleavage sequence (ENLYFQS). FXC TIM designs were produced as fusions to a C-terminal 6xHis tag. Expression was performed in E. coli BL21(DE3) using the pET24a expression vector under an isopropyl β-D-thiogalactopyranoside (IPTG) inducible promoter. Cultures were induced at OD600 0.5–0.8 by 1mM IPTG at 16 °C for 18–20 h. Proteins were purified by Ni-NTA-affinity resin (Qiagen). Purity and monomeric state were confirmed using a Superdex 75 increase 10/300 GL column (GE Healthcare) and SDS-PAGE gels. Size exclusion chromatography (SEC) data reported is immediately post-Ni-NTA purification without additional purification steps in either Phosphate Buffered Saline (PBS) at pH 7.4 or in 50 mM MES, 50 mM NaCl buffer at pH 6.0. Sample-specific buffers and protein concentrations ahead of SEC are reported at https://drive.google.com/file/d/1lyTwBMm72GpN_qVWLvoK2dMChYuazOTR/view?usp=sharing. Protein concentration was determined using predicted extinction coefficients and 280 nm absorbance measured on a NanoDrop spectrometer (Thermo Scientific).

**Circular dichroism spectroscopy**. Circular dichroism spectra were collected using a Jasco 815 spectropolarimeter with all measurements taken in Phosphate Buffered Saline (PBS) at pH 7.4 using a 1.0 mm path length cuvette. Cleanest post-SEC fraction(s) with sufficient protein were used for CD measurements. Wavelength scans were collected and averaged over 3 accumulations. Melting curves were collected monitoring CD signal at 222 nm over a range of 25 to 95 °C at 1 °C intervals, 1 min equilibration time and 10 s integration time. For 3mx7 designs, CD signal was monitored at 217 nm. Spectra are normalized to mean residue ellipticity ($10^3$ deg cm$^2$ dmol$^{-1}$) from millidegrees, using cuvette path length, protein length including tags, and concentration measurement from 280 nm absorbance as described in the previous section. Melting temperatures were determined by fitting a sigmoid function to melting curves.

**Crystallography**. F2N, F2C, and F15C were prepared in a 50 mM MES 50 mM NaCl pH 6.0 buffer at the following concentrations as measured by 280 nm absorbance: F2N (23 mg mL$^{-1}$), F2C (23 mg mL$^{-1}$), and F15C (15.6 mg mL$^{-1}$). Samples used for crystallography were from Ni-NTA and SEC fractions, without additional purification. Crystallization trials were done by screening for crystallization conditions using INDEX (HR2-134), Crystal Screen HT (HR2-130), BCS screen (Molecular Dimensions), MemGold HT-96 (Molecular Dimensions), and a 96-well buffer-selective crystal screen developed at SSRL. A number of crystal hits came from the SSRL screen using Bis Tris (pH 6.0) as the buffer. The best diffracting crystals for F2C were from a well solution consisting of 25% PEG 3350, 0.150 M Li$_2$SO$_4$ H$_2$O, and 0.1 M Bis Tris (pH 6.0). The F15C crystals were from a well solution consisting of 30% PEG 3350, 0.150 M Mg(OAc)$_2$ 4H$_2$O, 0.1 Bis Tris (pH 6.0). The F2N crystals were from a well solution consisting of 25% PEG 3350, 0.15 M Mg(OAC)$_2$, 0.1 M Bis Tris (pH 6.0). The final crystal drops were setup manually as sitting drops in 3 drop crystallization plate that holds 40 μL well solution with each drop consisting of 1 μL of protein and 1 μL of well solution. The F2C crystals appeared after 2 days and the F2N and F15C crystals appeared after 3–4 days. Diffraction data was collected at 100 K using Stanford Synchrotron Radiation Lightsource (SSRL) beamlines 12-2 and the Pilatus 6M detector. Data were indexed and integrated using XDS package[71]. The criteria used for high-resolution cutoff for the diffraction data are I/sigma ≥1.5 and CC(1/2) ≥70%. The sigma cutoff used for the F2N data is 1.4 because of the high redundancy (33) and high CC(1/2) value (82%). Initial phases were obtained by molecular replacement by using the program Phaser[72] and the coordinates of the designed structures as

the search model. The high-resolution F2C structure was traced with Buccaneer[73] and went through manual fitting using COOT[74] and refinement using REFMAC[75]. The F15C structure refinement involved several cycles of manual model building and refinement. The refinement statistics are provided in Supplementary Table 30.

**Reporting summary**. Further information on research design is available in the Nature Research Reporting Summary linked to this article.

## Data availability

Data generated and analyzed in this paper are available at this link: https://drive.google.com/drive/folders/1gBfu5LG8-kp9o7qBMkCdBSRCd0E8R6Te?usp=sharing. Crystal structures have been deposited in the Protein Data Bank with accession codes 7MCC (F2C), 7MCD (F15C), and 7SMJ (F2N). All data generated in this study are included in the manuscript, supplementary data files, and links listed in our open-source repository. Training data is publicly available from the PDB (Protein Data Bank) but also provided in prepared form at this link: https://console.cloud.google.com/storage/browser/seq-des-data. Trained models are available at https://drive.google.com/file/d/1X66RLbaA2-qTlJLlG9T153cao8gaKnEt/view?usp=sharing. The following databases were used for multiple sequence alignments: UniRef90, UniRef100, UniProtKB/Swiss-Prot (SP).

## Code availability

Training data and code to train the model and run the method are available at https://github.com/ProteinDesignLab/protein_seq_des. Commands to reproduce all design runs are provided in Supplementary Data 2 and 3.

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

## Acknowledgements
We thank Tudor Achim for the detailed discussion and feedback. We thank Wen Torng, Sergey Ovchinnikov, and Alex E. Chu for helpful discussion and Frank DiMaio for providing a set of decoys from which we selected test case structures. We thank Rohan Kannan for help with running and analyzing rotamer repacking experiments. This project was supported by the U.S. Department of Energy, Office of Science, Office of Advanced Scientific Computing Research, Scientific Discovery through Advanced Computing (SciDAC) program. Additionally, cloud computing credits were provided by Google Cloud. R.B.A. acknowledges support from NIH GM102365, GM61374, and the Chan Zuckerberg Biohub. R.R.E acknowledges support from the Stanford ChEM-H Chemistry/Biology Interface Predoctoral Training Program and the National Institute of General Medical Sciences of the National Institutes of Health under Award Number T32GM120007. A.D. acknowledges support from the National Library of Medicine BD2K training grant LM012409. Use of the Stanford Synchrotron Radiation Lightsource, SLAC National Accelerator Laboratory, is supported by the U.S. Department of Energy, Office of Science, Office of Basic Energy Sciences under Contract No. DE-AC02-76SF00515. The SSRL Structural Molecular Biology Program is supported by the DOE Office of Biological and Environmental Research, and by the National Institutes of Health, National Institute of General Medical Sciences (P30GM133894). The contents of this publication are solely the responsibility of the authors and do not necessarily represent the official views of NIGMS or NIH.

## Author contributions

N.A. and P.-S.H. conceived the research. N.A. built the algorithm and created the designs with discussions and assistance from R.R.E. and A.D. C.P.P., N.A., and R.R.E. conducted experimental validation. I.I.M. solved the protein structures. R.R.E. built the pipelines involving Rosetta. A.D. trained an initial model and contributed to ab initio experiments with R.R.E. C.P.P., R.R.E., and A.D. contributed to discussion and analysis. P.-S.H. and R.B.A. supervised the research. All authors contributed to the writing and editing of the manuscript.

## Competing interests

The authors declare no competing interests.
