## [Peer Review File · Nature Communications]

Protein sequence design with a learned potentialReviewers' Comments:

Reviewer #1:

Remarks to the Author:

The paper by Anand-Achim and colleagues describes a new method based in machine learning techniques to perform structure-based computational design. The computational design of new sequences starting from structural templates remains an unsolved and complex problem as such the subject of the paper is timely and an important need for the large community of protein engineers and biochemists. The methodological approach is novel, well described and thoroughly compared to other well validated methods. The results are extremely relevant for the field and are presented in a very clear way. The results are also well supported by the data and the inclusion of a detailed computational benchmarks and a considerable amount of experimental data makes this work a very robust and significant study. The methodological approach is sound and the github repository is well documented suggesting that this will be a tool that it will be easy to use and accessible to other groups. I support the publication of the work after minor clarifications.

Specific comments:

I) On the data shown in figure 2 what is the sequence identity of the designs tested experimentally relatively to the native proteins? The same question for the designs showed in figure S11-S13. This information should be clearly stated in the manuscript maybe even providing a sequence alignment, and perhaps it is but I may just have missed it.

II) I would encourage the authors to enrich figure 3 with some of the data they have in supp material.

III) The model used to design the de novo TIM- barrels also included native TIM-barrel topologies. If one would remove all the native TIM-barrels from the training data – would the results be dramatically different? I think it is important to address such a question as it speaks to the capability of the network to generalize.

IV) The systems designed in this paper were either small proteins (Fig 2) or proteins with internal symmetry like the TIM barrel where I understood correctly one only needs to design a small part of it and then rely on symmetry to repeat the sequence across the different modules of the structure. What is the performance dependency on protein size for the use of this method, both in terms of sequence recovery and run time?

Reviewer #2:

Remarks to the Author:

Anand-Achim et al. developed a deep neural network-based model for structure-based protein design, benchmarked their methods through a set of test proteins that were topologically irrelevant to the training proteins, and finally validated a few designs through biophysical experiments such as CD and X-ray. The authors concluded that the model generalizes to native topologies to produce experimentally stable designs. Overall, the work is novel and interesting, and should be attractive to readers in the protein design community. I have the following comments:

1. The authors observed a few expected structural features across the test designs and they described that “by inspection, polar networks supporting loops and anchoring secondary structure elements”. Those shown in Fig S5 are just models, and I am not sure if these designed anchoring residues can indeed maintain the expected hydrogen-bonding networks. Can the authors solve the X-ray structures of these designs to provide some evidence? I understand that they have performed CD to examine the secondary structure contents but CD results may not match well with the 3D information.
2. The authors mentioned that they use the Rosetta AbInitio application to predict the structures of

selected designs. Now since the AlphaFold2 and RoseTTAFold have been released and recent benchmark results showed that these two protocols are much more accurate than the old Rosetta AbInitio, I wonder if the authors will predict and report the results with these new methods.

3. The authors have the resources to solve the 3D structures of designed sequences. I wonder if they could also solve the structures for the top designs of the test proteins (e.g., 1acf, 1bkr, 1cc8, and 3mx7) just like what they did to the de novo TIM.

4. The test set for the rotamer and sequence recovery test is too small compared to a few other computational studies. For the rotamer recovery test, they achieved an accuracy of 72.6% for all rotamer angles being predicted within 20 degrees of the native angle. This is much better than those methods tested in a recent study (doi:10.1093/bioinformatics/btaa234) using the same stringent criterion of 20 degrees instead of the previously widely used, very loose criterion of 40 degrees. However, the accuracy in this study was obtained on only a tiny set of 4 proteins. Therefore, it is unknown whether this high accuracy is attributed to the "smart" selection of the 4 proteins. I would suggest the authors test their model for rotamer recovery with a large data set and compared it with the other packers such as those tested in 10.1093/bioinformatics/btaa234.

5. The benchmark of sequence recovery test has the same problem due to the tiny test set. They obtained a sequence recapitulation rate of 25~45% for the four proteins. I would suggest the authors evaluate their model's performance on a much larger set of non-redundant proteins (e.g., the carefully selected 136 monomer proteins in doi:10.1021/acs.jcim.9b00812) and compare the large-scale benchmark with other methods such as Rosetta and EvoEF2.

6. The authors described that "... the model designed sequences do not have hydrophobic residues in solvent-exposed positions, likely due to the abundance of cytosolic protein structures available ...". However, as shown in Fig S7C, the designed sequences still have a few hydrophobic residues in the solvent-exposed positions. It may not be precise to use "do not have".

7. In the Figure legend of Figure S14, it is mentioned that "In the low RMS regime ($< 5-10 \text{ \AA}$), the model and Rosetta are able to rank low RMS structures to a similar extent." In figure S14E, most of the curves, if not all, show large differences between the model and Rosetta in 5-10 angstroms, thus the above description may not be precise.

Reviewer #3:

Remarks to the Author:

The manuscript submitted by Anand-Achim and cols describes an interesting approach incorporated to de novo protein design. They investigated a sound methodology using a deep neural network model to design new protein sequences. The novelty is that the model explicitly builds rotamers and evaluates entire atom structural models, designing some successful folded and stable proteins. I'm not an expert in machine learning methods, so my comments focus on the results derived from the method, assuming that this one was performed correctly and will be possibly assessed by other reviewers. I consider that even the paper sometimes is difficult to follow without the presence of subtitles, the work presents a convincing methodology/results to be published in Nature Communications, adding knowledge to the protein design field and expanding the examples in the area, which will be well received by the protein design community. However, I would recommend the paper after some modifications/questions are addressed, indicated as follows:

- The text should be separated with corresponding subtitles to help the reading. In the current version, even the story is nicely flowing, it would be so helpful to separate the text with different sections and subtitles in the results/discussion part.
- They tested the model performance to design new sequences for de novo structures using a TIM-barrel backbone. The model seems to be able to fully redesign sequences given a specific backbone. However, is this model able to suggest sequences of backbones not present in the training set? I mean, only using key structural elements seen in natural folded proteins. For example, creating sequences compatible with de novo folds like Top7 or others but without including these topologies in

the training set.

- On page 27, I understand the reason behind the assumption of identity and conformation of each side chain being independent of all other side chains. However, could the authors say if exist some limitations with the premise? This is because it has been shown that coupling residues sometimes is essential for stability/function/cooperativity in proteins.
- Regarding the classifier performance, NQ prediction seems to be a bit problematic since even is predicted in some way NQ residues, also there are other outputs with a similar percentage. Also, the accuracy appears to be lower for NQ. I think the authors don't discuss this issue and could be briefly commented on.
- In Figure 2, panel I, for the design of 1bkr, looks at a huge increase of CD signal at 208/222. Do the authors know why this happened? Is it an artifact of protein concentration in the measurement? For me, the difference is big between native and model.
- Also, in Figure 2, panels I and J, which of the four designs presented in figs S11-S13 are showed to compare Native/Model? Authors could indicate that either in the text or figure legend.
- Do authors know why the error bar size of native sequence recovery for core residues in 1cc8 is so big? (figure 2 panel D).
- Why in some cases the model assigns low negative PLL to high RMSD backbones like in 1cc8? It's not an essential point of the manuscript, but it looked interesting to me. Maybe I missed the explanation.
- Figure S6 is a bit confusing how it is organized, also its associated text on page 7. Could the authors explain what features they meant with "the model designs some features that are not seen in the native sequence"?
- The authors stated, "though the model is trained to learn sequence dependence on structure, it makes correct predictions about structures given sequence." Do they have any hypothesis/idea why it happened? For me, the result of this "inverse" dependence is quite interesting.
- Do authors follow a four-fold symmetry approach because all validated de novo TIM barrels so far reported have 2- or 4-fold symmetry? Or do exist any other reasons? Did they try to redesign TIM-barrel backbones but not following a four-fold symmetry?
- On page 9, the authors mentioned a "key asparagine residue required for the structure to cooperatively fold," but I think it is actually an aspartate that switched the structure from a molten globule to a cooperatively folded structure.
- In Figure S15, authors should report the funnel plots with the same scale (y-axis) for all panels to give an easy comparison and avoiding misinterpretations from the readers, e.g., F8 seems a good design but placing it in the same scale for the others, it's closer to the upper right corner than the others.
- Some values, particularly RMSD computed ones, are listed with different and very high precisions decimal places. For clarity, could be shown with the same significant digits.
- For clarity and easy reading, it would be good if authors write the names of the four cooperatively folded proteins (page 9, third paragraph), and readers don't have to go to see fig. S17.
- All the folded proteins designed by the model have higher thermal stability, is there any reason why

it happened?

- Figure S17. For F15C, it's unclear if the T_m value is around 84 °C as authors reported or a higher temperature since the endotherm doesn't show a clear cooperative transition. Although these values are not relevant for the conclusions, the authors should modify the corresponding text to state the possibility of non-cooperative transitions for F5N and F15C or transitions occurring >95 °C, which could be addressed by DSC.
- For me, it's not a minor artifact the no proper closure of the barrel. Authors argue that it's most likely due to tags either at N- or C-terminal, which could be true; however, also exists the possibility that some interactions or molecular strain in the construct don't allow proper closure of the barrel, especially for F15C. Trying to crystallize the barrel without any tag might help to address this issue, but since implying more work, and this structure is not the significant contribution of the work, I think authors at least should discuss the role of backbone strain as been discussed by others (e.g., Koga et al., 2021). Also, I'm not convinced that an SO4 interaction would displace all the C-terminal helix; more if this interaction seems to be non-specific, it could be that the SO4 group is found there not as a cause but for a consequence of the displacement.
- The mutations in the interfaces between helices are interesting. I'm curious if some disrupting interactions are present in F2C/F2N that promote the conformation change in the C-terminal helix besides the possibility of disruption promoted by the tag. Did the authors check them?
- I have some concerns with respect to TIM barrels structures and their correlation with SEC experiments. For F15C, analytical SEC suggests higher oligomeric states besides monomers that could explain the dimer of dimers in the asymmetric unit. However, it seems like the authors didn't consider the presence of a second peak. The same applies to many other descriptions when the authors stated that proteins elute as monomers, which is partially true. In the case of existing conformational flexibility in the loops, this is not a negative property like it might be interpreted with the current description. Also, some flexibility is important to maintain the stability of the barrel and could help to functionalize the barrel in the future.
- I acknowledge the great methodological details for the computational part, and it will be so helpful for readers and the community. However, for the experimental section, some details are missing, like purification buffers and pH, information for analytical SEC (at least should be reported the protein concentration and indicate if the sample comes directly after IMAC step or previously was done a preparative SEC), crystallographic data, among others.
- In Figure S16, I think upper labels should be placed below sTIM-11 panels; otherwise, it gives a wrong idea that also is reported the elution for sTIM-11 with N- and C- terminal His-tag/His-TEV tag. Authors should add some discussion about this figure where indicate that elution profiles show multiple peaks and multiple oligomeric states in the samples (at least for F5N, F2C, F5C, among others). I assume that the indicated MW in each panel is the theoretical MW from the sequence, so it should be stated. The results for F2C and F15C are interesting since they are crystallized designs. Although the authors discuss the structural issues found in F2N structure, I think they have to discuss the possibility of that oligomeric state also happen in solution, not as stated now "although the protein elutes as a monomer in size exclusion chromatography" (first lines page 10), an argument which I don't completely share just looking the elution profiles.
- Related to the previous comment, in the supplementary text regarding crystal structure data of F2, the authors stated: "Both F2C and F2N are monomeric off of SEC", which is almost valid for F2N but not for F2C where clearly more oligomeric/dynamic states are present in the sample. Two paragraphs below, the authors wrote "For F2C, we see crystal contacts between the C-terminal helix and an adjacent monomer in the crystal (Fig. S18D); however, the protein is monomeric in solution based on SEC data (Fig. S16)", again this not completely valid looking the corresponding panel in Fig. S16.

Therefore, the multiple peaks could explain the possibility that some contacts between monomers observed in the structure are also present in solution, and the authors should rephrase these sentences.

- Continuing with the SEC profiles/structure discussion but now of F15C construct, authors stated on page 43: "We do not believe this cross-chain interaction is indicative of what happens in solution, as the protein is monomeric off of SEC (Fig S16)." Like the previous comments, authors should rephrase the corresponding sentences stating the multiple peaks in the analytical SEC elution and the presence of multiple oligomeric states, which is remarkably present on the crystallized designs, F2C and F15C.
- B-factors for the structure look high, a bit expected at the closure of the barrel as been observed for sTIM11 and DeNovoTIMs. However, it is surprising the vast difference between F2C and F2N when the only difference is the location of the same tag (fig. S18B). Is that an artifact from refinement strategy/or coloring scale? I think authors should discuss something related to this considerable difference.
- Could authors add a label that indicates the up panels on Fig. S19A are the top view of the barrels, and below is the bottom one? It would be helpful for readers.
- Crystallographic parameters for F2N are missing (table S5).
- All designed sequences for TIM barrels and other topologies have to be reported (even the non-well behaved designs). In table S4 there are only some sequences reported.

Some minor issues:

- What do the authors mean by learned degeneracy? Do they refer to the ability of elements structurally different but with the ability to yield the same output?
- The baseline model accuracy of 33.5% is a standard/normal value? Maybe it could be briefly commented on for non-expert readers.
- Even the approach is promising, how can the authors anticipate it could be used to design water-mediated contacts, interfaces, etc.? Or did they mean using a different training set?
- Reference for NovoTIMs should be updated to the current publication (JMB, 2021). Also, names changed from NovoTIMs to DeNovoTIMs and can be updated in this manuscript to consistency.
- Why is reported NovoTIM14a and b?
- Protein purification, italics in *E. coli*.
- In methods, CD melting curves interval is from 25-90 °C even the plots are reported until 95 °C.
- In the crystallography methods, criteria for resolution cutoff have to be mentioned. Also, the PDB ID for F2N is missing; it's just indicated XXX.
- In the crystallographic table (table S5), some superscripts are missing on the table footnote. In addition, resolution limits for the outer shell should be reported.
- Why in figure S11, left panel for 1acf, the line indicates elution volume around 14 mL and not more to the right that matches with the elution of native protein? The same applies to the right panel (3mx7).

We appreciate the thoughtful insights provided by the reviewers and have made revisions in accordance with their suggestions. Our point-by-point responses are given below in red.

REVIEWER COMMENTS

Reviewer #1 (Expertise: protein design, computational methods):

The paper by Anand-Achim and colleagues describes a new method based in machine learning techniques to perform structure-based computational design. The computational design of new sequences starting from structural templates remains an unsolved and complex problem as such the subject of the paper is timely and an important need for the large community of protein engineers and biochemists. The methodological approach is novel, well described and thoroughly compared to other well validated methods. The results are extremely relevant for the field and are presented in a very clear way. The results are also well supported by the data and the inclusion of a detailed computational benchmarks and a considerable amount of experimental data makes this work a very robust and significant study. The methodological approach is sound and the github repository is well documented suggesting that this will be a tool that it will be easy to use and accessible to other groups. I support the publication of the work after minor clarifications.

Specific comments:

I) On the data shown in figure 2 what is the sequence identity of the designs tested experimentally relatively to the native proteins? The same question for the designs showed in figure S11-S13. This information should be clearly stated in the manuscript maybe even providing a sequence alignment, and perhaps it is but I may just have missed it.

We have updated the main Figure 2 and Figures S11-S13 with the sequence identity information and also provide new Supplementary Tables S2-S5 with the native test case design sequence alignments. We have also added sequence identity information to Figure 4 and Figures S15-S17 for the TIM-barrel designs. We note that we already report the native sequence identity in Figure S9 and provide all designed sequence data in Supplementary Datasets S2 and S3.

II) I would encourage the authors to enrich figure 3 with some of the data they have in supp material.

We have enriched Figure 3 as suggested by the reviewer.

III) The model used to design the de novo TIM- barrels also included native TIM-barrel topologies. If one would remove all the native TIM-barrels from the training data – would the results be dramatically different? I think it is important to address such a question as it speaks to the capability of the network to generalize.

Our goal in extensively characterizing designs on test set topologies was to address this exact point by the reviewer. We show that the model does in fact generalize to topologies not seen during training (Fig 2, Section 'Design algorithm generalizes to unseen backbone topologies'), and we do not believe that the results would be dramatically different as a result. Although the alpha/beta helical arrangement in the de novo TIM barrel is unique, there could be a benefit from training on similar but non-identical backbone structures.

IV) The systems designed in this paper were either small proteins (Fig 2) or proteins with internal symmetry like the TIM barrel where I understood correctly one only needs to design a small part of it and then rely on symmetry to repeat the sequence across the different modules of the structure.

For symmetric design, the design procedure is run on the full TIM-barrel structure, but the model-predicted logits at symmetric positions are averaged and only one residue type and rotamer is built at each symmetric position per step. In essence, the model is designing a single symmetric subunit and the two interfaces between the symmetric subunits as well as the inner core of the barrel.

What is the performance dependency on protein size for the use of this method, both in terms of sequence recovery and run time?

We provide runtime data in Supplementary Dataset S2. The exact run-time scaling, as we describe in the Methods section, will be a function of the number of GPUs used and their type. We use a fixed annealing schedule (number of iterations and temperature annealing) for all proteins, and it is possible that these

parameters might have to be modified for larger proteins (more iterations, slower annealing). However, with larger proteins the block size for blocked sampling will also increase, as there will be more residues that are conditionally independent of each other, reducing the total number of iterations needed.

We plot here the runtime data we provide in Supplementary Dataset S2 for design runs on one GPU. We note that the model runtime spread is due to running multiple and varying numbers design runs on the same machine and same GPU (effective batch size larger than the number of residues in each test case). As we state in the methods, the runtime is expected to improve significantly with multi-GPU parallelism. We are currently working on a followup study to improve the run-time. By providing initial guesses and changing annealing schedule we can speed up the algorithm by ~70% without significantly changing the output sequence. This work, however, is beyond the scope of this paper.

Runtime in hours across design runs done on a 32 GB RAM machine and on a single GeForce GTX TITAN X GPU for PDB templates. Note that some design runs were run in parallel, leading to worse runtime. Number of residues: 1acf (125), 1bkr (108), 1cc8 (72), 3mx7 (90)

We provide data below to show the native sequence recovery rate across all design runs as a function of the number of iterations, also noting that native sequence recovery alone, as we discuss in the paper, is not a singular metric of performance.

Native sequence recovery rate as a function of number of annealing iterations for each PDB backbone template, across 100 design runs.

Reviewer #2 (Expertise: protein design, computational methods):

Anand-Achim et al. developed a deep neural network-based model for structure-based protein design, benchmarked their methods through a set of test proteins that were topologically irrelevant to the training proteins, and finally validated a few designs through biophysical experiments such as CD and X-ray. The authors concluded that the model generalizes to native topologies to produce experimentally stable designs. Overall, the work is novel and interesting, and should be attractive to readers in the protein design community. I have the following comments:

1. The authors observed a few expected structural features across the test designs and they described that “by inspection, polar networks supporting loops and anchoring secondary structure elements”. Those shown in Fig S5 are just models, and I am not sure if these designed anchoring residues can indeed maintain the expected hydrogen-bonding networks. Can the authors solve the X-ray structures of these designs to provide some evidence? I understand that they have performed CD to examine the secondary structure contents but CD results may not match well with the 3D information.

The reviewer is correct that the data in Fig S5 are from model designs. We present these to show that the model is capable of suggesting and designing polar contacts/networks. We also note that the model can recapitulate polar contacts seen in the native crystal structures, which can be seen in the AI-designed models provided in Supplementary Dataset S4. The goal of this Figure is to show non-native, novel polar contacts/networks that the model suggests. We also note that the model designs novel polar contacts/networks for the TIM-barrel structure that we do in fact validate through to X-ray crystal structures (Figure 4).

We have attempted to solve the 3D X-ray crystallography structures for the native test case designs, all the way through multiple rounds of expression and purification to get high-purity, high-concentration samples, through to setting crystal trays starting in January 2021. Unfortunately, we have not been able to find crystal-forming conditions despite extensive testing. We have shown that a subset of the native protein designs have the expected CD signa, as well as cooperative unfolding transitions or hyperthermostability that suggest the designs are stably folded proteins (Figures 2, S12).

2. The authors mentioned that they use the Rosetta Ablnitio application to predict the structures of selected designs. Now since the AlphaFold2 and RoseTTAFold have been released and recent benchmark results showed that these two protocols are much more accurate than the old Rosetta Ablnitio, I wonder if the authors will predict and report the results with these new methods.

We are exploring evaluating designed sequences with learned structure prediction methods in future work, and we considered including results from newer structure prediction methods. However, we think reporting these results in the context of this paper might be misleading for the following reasons. The learned structure prediction methods train on crystal structures in the PDB, likely including our test set structures and the crystal structure for the original TIM-barrel design *5bvl*. Moreover, we see that for the native test case designs there is enough sequence identity with the native such that hundreds of homologous sequences are found when doing database lookups for MSA, and we know that MSA features provide a rich statistical prior for predicting structural contacts. Even without MSA features, if there is enough signal to predict in effect secondary structure accurately from sequence (and we know Psipred predictions for instance are accurate), predicting structures from the train set would then be trivial for these over-parameterized models. For these reasons, we do not think learned structure prediction methods are an unbiased way to evaluate designs. For completeness, however, we do include data for the de novo TIM-barrel structure predictions from trRosetta (Figure S10), as the structure recovery component of the predictor is not learned, which might counteract the bias from training on the ground-truth crystal structures.

3. The authors have the resources to solve the 3D structures of designed sequences. I wonder if they could also solve the structures for the top designs of the test proteins (e.g., 1acf, 1bkr, 1cc8, and 3mx7) just like what they did to the de novo TIM.

As mentioned, we have attempted to solve the 3D X-ray crystallography structures for the native test case designs, all the way through multiple rounds of expression and purification to get high-purity high-concentration samples, through to setting crystal trays starting in January 2021. Unfortunately, we have not been able to get high-resolution crystal structures for these sequences. We believe the extensive computational metrics and the CD and SEC data provided indicate that the model can produce designs that have expected features, that express, are folded and soluble, and that have the correct secondary structure signal.

4. The test set for the rotamer and sequence recovery test is too small compared to a few other computational studies. For the rotamer recovery test, they achieved an accuracy of 72.6% for all rotamer angles being predicted within 20 degrees of the native angle. This is much better than those methods tested in a recent study (doi:10.1093/bioinformatics/btaa234) using the same stringent criterion of 20 degrees instead of the previously widely used, very loose criterion of 40 degrees. However, the accuracy in this study was obtained on only a tiny set of 4 proteins. Therefore, it is unknown whether this high accuracy is attributed to the “smart” selection of the 4 proteins. I would suggest the authors test their model for rotamer recovery with a large data set and compared it with the other packers such as those tested in 10.1093/bioinformatics/btaa234.

Although we do present a learned method for sequence design, our focus in this study was not to do extensive benchmarking, but rather to explore what this type of model can do, what it learns, and to evaluate generated sequences through experiments. In addition, our focus in this study is on sequence design and the rotamer repacking results are presented for completeness. We are looking into exploring this type of benchmarking in future work.

As our model is based on training on native crystal structures, it can potentially memorize data. To evaluate model generalization, we remove entire topology classes from the train set to make our test cases unbiased. A proper dataset equivalent to the benchmarks used for the packer, which covers a large number of topologies and fold instances, is not straightforwardly manageable without removing a large swathe of data and possibly crippling the training of this model. To properly address this issue, we need more de novo protein test cases with validated structures. We are indeed working on this with other protein design groups for a followup study.

Finally, as discussed in the text, the test case structures were selected to span a range of CATH classes. One point we have added in the main text is that the test case structures were selected because they were validatable with Rosetta ab initio forward folding. As we use ab initio folding as a metric for evaluating designs, it was necessary to select test case structures that could be recovered by Rosetta forward folding. Moreover, we do not present the accuracy data as benchmarked across a large dataset, focusing instead on evaluating few structures deeply -- across tens of computational metrics and through to experimental validation.

5. The benchmark of sequence recovery test has the same problem due to the tiny test set. They obtained a sequence recapitulation rate of 25~45% for the four proteins. I would suggest the authors evaluate their model's performance on a much larger set of non-redundant proteins (e.g., the carefully selected 136 monomer proteins in doi:10.1021/acs.jcim.9b00812) and compare the large-scale benchmark with other methods such as Rosetta and EvoEF2.

As discussed above, although we do present a method for sequence design, our focus in this study was not to do benchmarking, but rather to explore the extent to which a learned method could design sequence and to investigate the quality and viability of AI-generated designs (although we do extensively characterize designs and compare head-to-head with Rosetta in the supplement). We are looking into exploring this type of larger-scale evaluation in future work, as it is definitely of interest. Another subtle point we make in the paper is that native sequence recovery is an informative but certainly not sufficient metric for evaluating designed sequences for a given backbone. For example, if the method recovered native sequences with 90+% accuracy, it would indicate that the model had (1) seen the backbone during training and (2) overfit/memorized the native sequence. We evaluate the model designs under a range of metrics in order to demonstrate the viability of the designs beyond native sequence recovery.

6. The authors described that "... the model designed sequences do not have hydrophobic residues in solvent-exposed positions, likely due to the abundance of cytosolic protein structures available ...". However, as shown in Fig S7C, the designed sequences still have a few hydrophobic residues in the solvent-exposed positions. It may not be precise to use "do not have".

We have modified the text as suggested.

7. In the Figure legend of Figure S14, it is mentioned that "In the low RMS regime (< 5-10 Å), the model and Rosetta are able to rank low RMS structures to a similar extent." In figure S14E, most of the curves, if not all, show large differences between the model and Rosetta in 5-10 angstroms, thus the above description may not be precise.

We have modified the text to be more precise. For some structures (1bkr, 3mx7) the model ranks decoys better (1bkr) or on par with Rosetta (3mx7) in the 5-10 angstrom regime; however, we agree that it is more precise to change the language here.

Reviewer #3 (Expertise: protein design, protein folding, experimental):

The manuscript submitted by Anand-Achim and cols describes an interesting approach incorporated to de novo protein design. They investigated a sound methodology using a deep neural network model to design new protein sequences. The novelty is that the model explicitly builds rotamers and evaluates entire atom structural models, designing some successful folded and stable proteins. I'm not an expert in machine learning methods, so my comments focus on the results derived from the method, assuming that this one was performed correctly and will be possibly assessed by other reviewers. I consider that even the paper sometimes is difficult to follow without the presence of subtitles, the work presents a convincing methodology/results to be published in Nature Communications, adding knowledge to the protein design field and expanding the examples in the area, which will be well received by the protein design community. However, I would recommend the paper after some modifications/questions are addressed, indicated as follows:

• The text should be separated with corresponding subtitles to help the reading. In the current version, even the story is nicely flowing, it would be so helpful to separate the text with different sections and subtitles in the results/discussion part.

We have updated the text with subtitled sections as the reviewer suggests.

• They tested the model performance to design new sequences for de novo structures using a TIM-barrel backbone. The model seems to be able to fully redesign sequences given a specific backbone. However, is this model able to suggest sequences of backbones not present in the training set? I mean, only using key structural elements seen in natural folded proteins. For example, creating sequences compatible with de novo folds like Top7 or others but without including these topologies in the training set.

Yes, the model suggests sequences for backbones not present in the training set; this is what we show when we evaluate designs on native test case structures that are stringently split from the train set at the CATH topology level (Fig 2).

We believe our empirical results indicate that the model would generate to *de novo* backbones with topologies not seen during training. This is because

1. The test structures are separated stringently at the CATH topology level, and the model is able to design sequences for these non-training set topologies. This implies that the model has learned to identify key structural elements that recur across protein topologies and are not unique to the train set structures.
2. The model is able to generalize to a de-novo designed TIM-barrel backbone design model, although it is trained on crystal structures, suggesting that it should generalize to de novo protein backbone models.

• On page 27, I understand the reason behind the assumption of identity and conformation of each side chain being independent of all other side chains. However, could the authors say if exist some limitations with the premise? This is because it has been shown that coupling residues sometimes is essential for stability/function/cooperativity in proteins.

The assumption is more specifically that the identity and conformation of a side chain *conditioned on neighboring side chains* is independent of more distal residues that do not neighbor it. The algorithm does in fact capture residue coupling -- but this is via the sampling procedure. Capturing residue coupling (for a given backbone) is how our model achieved the design results we observed, such as packing, hydrogen bonding, etc. The assumption the reviewer describes motivates the classification task -- predicting residue identity conditioned on the surrounding local chemical context.

• Regarding the classifier performance, NQ prediction seems to be a bit problematic since even is predicted in some way NQ residues, also there are other outputs with a similar percentage. Also, the accuracy appears to be lower for NQ. I think the authors don't discuss this issue and could be briefly commented on.

We have added some discussion about the classifier performance to the Supplementary Text. Lower accuracy for Q prediction is likely a function of lower Q abundance (see Fig S4B) as well as inherent difficulty in predicting surface polar residues without sufficient context. This could be combated in future with resampling or class weighting during training, which should improve model accuracy for predicting Q in particular.

• In Figure 2, panel I, for the design of 1bkr, looks at a huge increase of CD signal at 208/222. Do the authors know why this happened? Is it an artifact of protein concentration in the measurement? For me, the difference is big between native and model.

Protein concentrations were measured via A280 absorption on a NanoDrop machine using predicted extinction coefficients to calculate the final concentration. As the reviewer suggests, the difference in the normalized CD signal magnitude for 1bkr d2 compared to the native 1bkr sequence could be due to errors in protein concentration determination by A280 absorption. We have updated the Methods section to clearly state that the protein concentration is determined in this manner and used to normalize the CD spectra. Our goal in presenting this data was (1) to show that the proteins are folded and (2) to show that the proteins have the expected secondary structure signal under CD. For the design in question, the CD data clearly shows an all-alpha protein fold.

- Also, in Figure 2, panels I and J, which of the four designs presented in figs S11-S13 are showed to compare Native/Model? Authors could indicate that either in the text or figure legend.

We have updated Figure 2 to include this information.

- Do authors know why the error bar size of native sequence recovery for core residues in 1cc8 is so big? (figure 2 panel D).

The standard deviation of core native sequence recovery is high because some model designs almost exactly recapitulate the core residues, and others are alternate designs.

- Why in some cases the model assigns low negative PLL to high RMSD backbones like in 1cc8? It's not an essential point of the manuscript, but it looked interesting to me. Maybe I missed the explanation.

We point out in the legend for Figure 3 that the low negative PLL structure for 1cc8 has an "alternative pattern of beta strand pairing." We have updated the main text to include this and other observations.

The negative PLL is an approximation of the joint probability of the amino acids/rotamers given the backbone. So, very simply, if the probabilities of side-chains conditioned on local chemical context is high, the negative PLL will be low. In essence, this happens when the model is confident about a majority of side-chain identities and their rotamers across the protein. The alternative beta strand pairing structure still forms a very similar alpha-beta topology where the core residues are still in core, buried positions, with polar residues exposed, so perhaps it is reasonable that the negative PLL heuristic pulls out this structure.

- Figure S6 is a bit confusing how it is organized, also its associated text on page 7. Could the authors explain what features they meant with "the model designs some features that are not seen in the native sequence"?

Here we point out that the model designs might differ from the native sequence, yet converge on sequence features seen in homologous sequences that align to the native sequence in MSA lookups. The figure highlights sequence positions where the designed residue differs from the native sequence but appears in a homologous sequence (reported in text above each panel).

The current text is "Interestingly, the model designs some features that are not seen in the native sequence, yet appear in homologous sequences (Fig S6)." We have updated this text to be more clear.

- The authors stated, "though the model is trained to learn sequence dependence on structure, it makes correct predictions about structures given sequence." Do they have any hypothesis/idea why it happened? For me, the result of this "inverse" dependence is quite interesting.

We touched on this in our response to the question earlier about 1cc8 but will elaborate further. We think that for unlikely backbone configurations for a given sequence, the local chemical context around individual residues will be out-of-distribution for the model, and empirically we see that the side chains' probabilities conditioned on local context are low under the model. In contrast, for a predicted structure with reasonable backbone and side-chain configurations such that each residue is likely under the model conditioned on the residue's neighboring chemical context -- the model negative PLL will be low. Therefore, the model negative PLL seems to function as a good heuristic for how likely a structure is for a given sequence.

In general, though, the model is worse at decoy ranking for higher-deviation backbones, as the model lacks is an ability to 'scale' the negative PLL score; that is, if one or a few residue positions are low probability, that will not necessarily scale the score which is just a sum of the negative log probabilities. This is why we see that the model in some cases scores a 5 angstrom CA RMSD decoy similarly to a 10+ angstrom CA RMSD decoy.

- Do authors follow a four-fold symmetry approach because all validated de novo TIM barrels so far reported have 2- or 4-fold symmetry? Or do exist any other reasons? Did they try to redesign TIM-barrel backbones but not following a four-fold symmetry?

Yes, we design with four-fold symmetry in order to compare the designs with previous structure-confirmed four-fold symmetric designs. We are pursuing experiments to redesign the TIM-barrel backbones without symmetry and will explore this in future work.

- On page 9, the authors mentioned a “key asparagine residue required for the structure to cooperatively fold,” but I think it is actually an aspartate that switched the structure from a molten globule to a cooperatively folded structure.

Corrected, as suggested.

- In Figure S15, authors should report the funnel plots with the same scale (y-axis) for all panels to give an easy comparison and avoiding misinterpretations from the readers, e.g., F8 seems a good design but placing it in the same scale for the others, it's closer to the upper right corner than the others.

Updated, as suggested.

- Some values, particularly RMSD computed ones, are listed with different and very high precisions decimal places. For clarity, could be shown with the same significant digits.

Updated in Figure S9 and S15, as suggested.

- For clarity and easy reading, it would be good if authors write the names of the four cooperatively folded proteins (page 9, third paragraph), and readers don't have to go to see fig. S17.

Updated, as suggested.

- All the folded proteins designed by the model have higher thermal stability, is there any reason why it happened?

This is certainly an interesting phenomenon. One explanation might be that our optimization procedure 'over-optimizes' the model negative PLL past the range of native model negative PLLs (see Figure S8C), and that procedure of producing designs that the model is more 'certain' about also leads to more thermally stable designs; for example, designs might have very few exposed hydrophobic residues as a result of these being penalized under the model, which might contribute to overall design stability. In Rocklin et. al., (DOI: 10.1126/science.aan0693) they observed that burial of hydrophobic surface area is a main driving design parameter for stability in designed proteins. Since the model is trained primarily on cytosolic proteins, it learns a bias toward burying hydrophobic residues and designing polar residues at solvent-exposed position, and we do see that optimizing the PLL under the model leads to reduced exposed hydrophobic residues in designs. More deeply understanding the reasons behind higher thermal stability of model designs will require future investigation.

- Figure S17. For F15C, it's unclear if the T_m value is around 84 °C as authors reported or a higher temperature since the endotherm doesn't show a clear cooperative transition. Although these values are not relevant for the conclusions, the authors should modify the corresponding text to state the possibility of non-cooperative transitions for F5N and F15C or transitions occurring >95 °C, which could be addressed by DSC.

Thank you for the detailed analysis here. The order of plots in the middle column for Figure S17 is incorrect. F15C does in fact have a clear cooperative transition. F5N and F5C both do not display a cooperative transition below 95 °C. We have updated Figure S17. The T_m value is 84.2 °C as reported.

- For me, it's not a minor artifact the no proper closure of the barrel. Authors argue that it's most likely due to tags either at N- or C-terminal, which could be true; however, also exists the possibility that some interactions or molecular strain in the construct don't allow proper closure of the barrel, especially for F15C. Trying to crystallize the barrel without any tag might help to address this issue, but since implying more work, and this structure is not the significant contribution of the work, I think authors at least should discuss the role of backbone strain as been discussed by others (e.g., Koga et al., 2021). Also, I'm not convinced that an SO4 interaction would displace all the C-terminal helix; more if this interaction seems to be non-specific, it could be that the SO4 group is found there not as a cause but for a consequence of the displacement.

To clarify, F15C folds correctly to the TIM-barrel structure. F2C/F2N have a dislodged C-terminal helix, although the barrel still closes fully (proper closure of the barrel).

However, we agree with the reviewer's point that there may be other possibilities and subtlety behind the C-terminal helix dislodgement. We include data from the structure with an N-terminal tag with the tag partially resolved in the density to show how the tag might disrupt the C-terminal helix placement in the crystal (Figure S18C). We have added more discussion around this structure in the main text and supplement. One point we have added is discussion around the Ala3 designed residue (and the equivalent 49, 95, 141 positions). Compared to F15 and 5bvl which have leucines designed at this position, it seems that this alanine in the core reduces the packing density. It appears that using alanine at this position still produces a stable design, but the terminal element carrying it was destabilized sufficiently to allow alternative arrangements of the secondary structure in the F2C and F2N crystal conditions.

Overall, since F15C folds as expected very nearly to the design template, we think the approach is valid, even if the F2C/N structures have a dislodged C-terminal helix.

• The mutations in the interfaces between helices are interesting. I'm curious if some disrupting interactions are present in F2C/F2N that promote the conformation change in the C-terminal helix besides the possibility of disruption promoted by the tag. Did the authors check them?

Yes, as we stated above, we have investigated the structure more closely and believe that design of Ala3 (the equivalent Ala49, Ala95, Ala141) reduces the hydrophobic volume in the core, and we think this allows other interactions in the crystallization conditions to dislodge the helix. We have included this discussion in the text and supplement.

• I have some concerns with respect to TIM barrels structures and their correlation with SEC experiments. For F15C, analytical SEC suggests higher oligomeric states besides monomers that could explain the dimer of dimers in the asymmetric unit. However, it seems like the authors didn't consider the presence of a second peak. The same applies to many other descriptions when the authors stated that proteins elute as monomers, which is partially true.

Responding to this and other comments on the oligomeric states of the protein further below.

In the case of existing conformational flexibility in the loops, this is not a negative property like it might be interpreted with the current description. Also, some flexibility is important to maintain the stability of the barrel and could help to functionalize the barrel in the future.

We did not intend to frame conformational flexibility in a negative light. In fact, we view conformational flexibility of the protein as a feature, not a bug. One of our stated goals was to see if the network could behave as a 'soft potential,' designing sequences that implicitly capture backbone flexibility, which Rosetta does not do well. As we say on pg. 3:

"We hypothesized that a learned model could operate as a "soft" potential that implicitly captures backbone flexibility, producing diverse sequences for a fixed protein backbone."

We have revised the supplementary text describing conformational flexibility based on the reviewer's comment to reflect this interpretation.

• I acknowledge the great methodological details for the computational part, and it will be so helpful for readers and the community. However, for the experimental section, some details are missing, like purification buffers and pH, information for analytical SEC (at least should be reported the protein concentration and indicate if the sample comes directly after IMAC step or previously was done a preparative SEC), crystallographic data, among others.

We agree that this data is critical to include and found that some of this data was mistakenly omitted from Supplementary Dataset S4. We have updated that dataset to include buffer composition and pH for SEC and protein concentration for CD. We have updated the Methods section to include the concentration of proteins ahead of crystallography. All CD data was taken in 1x PBS, pH 7.4. All proteins were purified via Ni-NTA

column purification with SEC immediately following before taking CD data and then concentrating down pure fractions for crystallography. We explored ion exchange for some of the TIM-barrel constructs but did not end up using samples post-ion exchange for crystallography. This information is now included in the Methods section, with buffer conditions and protein concentrations added to Supplementary Dataset S4.

• In Figure S16, I think upper labels should be placed below sTIM-11 panels; otherwise, it gives a wrong idea that also is reported the elution for sTIM-11 with N- and C- terminal His-tag/His-TEV tag.

Updated, as suggested

I assume that the indicated MW in each panel is the theoretical MW from the sequence, so it should be stated.

Updated, as suggested, for Figures S11 and S16.

Reviewer comments about the oligomeric state of the TIM-barrel proteins:

(Comment from earlier) • I have some concerns with respect to TIM barrels structures and their correlation with SEC experiments. For F15C, analytical SEC suggests higher oligomeric states besides monomers that could explain the dimer of dimers in the asymmetric unit. However, it seems like the authors didn't consider the presence of a second peak. The same applies to many other descriptions when the authors stated that proteins elute as monomers, which is partially true.

We think the reviewer definitely has a valid and insightful point here, and we are happy to clarify. For F15C, we did consider the second peak and present more data here to elucidate our thoughts on this molecule.

Our SEC runs are done immediately post-NiNTA, so there are other background contaminants present apart from the protein. We have clarified this in the text now. It might be helpful to look first at the data for F15N (below, same construct with N-terminal His-TEV tag instead of C-terminal His tag), which we investigated in more detail. Note that we were not able to get crystals for this particular construct. (For F15C, we did crystallography prep right after Ni-NTA and SEC purification and were successful, despite slight background contaminants).

(Left) SEC data for F15N (Superdex 75, 1xPBS), post-NiNTA. (Right) SEC fractions corresponding to left and right peaks run on denaturing gel.

For this protein, the monomer peak elutes as expected around 12.8 mL, with a left peak at 11.3 mL with a shoulder peak at 10.3 mL. However, after anion exchange on this sample and running pooled, clean fractions on SEC, the left peaks are no longer present (see data below).

(Right) SEC data for F15N (Superdex 75, 10mM Tris 50mM NaCl pH 8), post- anion exchange, cleanest fractions pooled. Peak at 12.5 mL. (Left) Anion exchange fractions for F15N, with clean fractions for SEC labeled.

The concentration is lower, but we see a single monodispersed monomer peak and no trace of oligomeric equilibrium. Therefore, we believe that at lower concentrations the protein is a monomer in solution. We did not do anion exchange for F15C, as we were trying to preserve as much sample as possible ahead of crystallography.

For F15C, the contacts between the monomers in the crystal structure are loose and highly solvated with polar residues. It lacks support to suggest that the proteins would necessarily be stably dimeric. We did not investigate the fractions containing background or soluble aggregates extremely thoroughly, and we were lucky to have been able to crystallize from a crude purification protocol. We believe that the left peak in the SEC data (below) is likely not indicative of dimer species but rather contaminants that we excluded from crystallization samples.

(Left) SEC data for F15C (Superdex 75, 50mM MES 50mM NaCl pH 6.0), post-NiNTA. (Right) SEC fractions corresponding to main peak. Majority of protein is in main peak. Cleanest fraction (rightmost) used successfully for crystallography. Fractions for left peak were not run on the gel.

The left peak is at 10.3 mL and the main peak is at 12 mL. Both peaks elute earlier than expected, with the dimer peak expected at around 12.5-13 mL. The dimer of dimers (perhaps corresponding to the 4 monomer asymmetric unit) would be expected to elute earlier than 9 mL. The dimer peak (~42 kDa) would be expected at just past 11 mL while the monomer peak is expected at around 12.5-13 mL. Based on the gel, we see that the protein might be interacting with one or more lower molecular weight background proteins, and some of these contaminants might not be soluble or monomeric either. Since the dimer peak is slightly left-shifted relative to where it might be expected, the peak could be an amalgam of background contaminants with our protein or the protein interacting with other unrelated components. Based on this data, and the ion exchange data for the same protein with an alternate tag, we believe the protein is likely monomeric.

In summary -- we have modified the text to indicate that the F15C protein appears likely monomeric off of SEC. We have also added the ion exchange data for F15N to Fig S16. We have also clarified that the SEC data reported is taken immediately after Ni-NTA purification without additional purification steps. For the purposes of showing that the designed sequence adopts the TIM-barrel fold, we believe the data provided is clear.

Authors should add some discussion about this figure where indicate that elution profiles show multiple peaks and multiple oligomeric states in the samples (at least for F5N, F2C, F5C, among others). [...] The results for F2C and F15C are interesting since they are crystallized designs. Although the authors discuss the structural issues found in F2N structure, I think they have to discuss the possibility of that oligomeric state also happen in solution, not as stated now "although the protein elutes as a monomer in size exclusion chromatography" (first lines page 10), an argument which I don't completely share just looking the elution profiles.

Our SEC runs are done immediately post-NiNTA, so there are other background contaminants present apart from the protein. For F2N, we do not resolve a cross-chain interaction in the crystal structure for F2N. For the SEC data post-NiNTA (data below), a small left peak elutes at 10.6 mL (dimer peak expected past 11 mL) and contains background contaminants.

(Left) SEC data for F2N (Superdex 75, 1xPBS) after Ni-NTA purification. (Right) 1 mL elution fractions run on denaturing gel.

We present here data for F2N post cation exchange, where pure fractions were pooled down before SEC.

(Right, top) SEC data for F2N (Superdex 75, 50mM MES 50mM NaCl pH 6), post- cation exchange, cleanest fractions pooled. Peak at 13 mL. (Left, top) Cation exchange fractions for F2N, with clean fractions for SEC labeled. (Below) SEC fraction for main peak in top plot.

We see here that F2N is clearly monomeric, although this data is for protein at around 5X lower concentration than the data for the protein immediately post-NiNTA.

Although we did not do ion exchange for F2C and proceeded straight to crystallography with main peak fractions off of post-NiNTA SEC (data below from two runs), the left-most SEC peak seen is at 9.8 mL which is just past the void volume and much earlier than where the dimer peak is expected (at after 11mL). and we do think the data shows that the structure is largely monomeric, with the main peak just before 13 mL, as seen with the sTIM11 protein.

(Left) SEC data for F2C (Superdex 75, 50mM MES, 50mM NaCl pH 6.0) after Ni-NTA purification. (Right) 1 mL elution fractions for main peak run on denaturing gel. Protein elutes in a broad peak with peak at 13 mL. The cleanest rightmost fraction was concentrated down ahead of setting crystal trays -- and we successfully got a structure from this sample despite the background low MW proteins.

(Left) SEC data for F2C (Superdex 75, 50mM MES, 50mM NaCl pH 6.0) after Ni-NTA purification (Units: mAU vs. mL). (Right) 1 mL elution fractions for main peak run on denaturing gel). Protein elutes in a broad peak.

In addition, the cross-monomer crystal contacts in the structure F2C do not suggest intimately associated dimer interactions and could be driven by the high local concentration within the crystal.

Overall, for these two constructs, we think it is fair to say the proteins are largely monomeric, and we have modified the text to indicate this. We have also added the ion exchange data for F2N to Fig S16. We have also clarified that the SEC data reported is taken immediately after Ni-NTA purification without additional purification steps.

- Related to the previous comment, in the supplementary text regarding crystal structure data of F2, the authors stated: “Both F2C and F2N are monomeric off of SEC”, which is almost valid for F2N but not for F2C where clearly more oligomeric/dynamic states are present in the sample. Two paragraphs below, the authors wrote “For F2C, we see crystal contacts between the C-terminal helix and an adjacent monomer in the crystal (Fig. S18D); however, the protein is monomeric in solution based on SEC data (Fig. S16)”, again this not completely valid looking the corresponding panel in Fig. S16. Therefore, the multiple peaks could explain the possibility that some contacts between monomers observed in the structure are also present in solution, and the authors should rephrase these sentences.
- Continuing with the SEC profiles/structure discussion but now of F15C construct, authors stated on page 43: “We do not believe this cross-chain interaction is indicative of what happens in solution, as the protein is monomeric off of SEC (Fig S16).” Like the previous comments, authors should rephrase the corresponding sentences stating the multiple peaks in the analytical SEC elution and the presence of multiple oligomeric states, which is remarkably present on the crystallized designs, F2C and F15C.

We have rephrased and clarified these sentences in line with the data provided above. We have also added more discussion in the main text and supplementary text around the proteins' oligomeric states in solution.

- B-factors for the structure look high, a bit expected at the closure of the barrel as been observed for sTIM11 and DeNovoTIMs. However, it is surprising the vast difference between F2C and F2N when the only difference is the location of the same tag (fig. S18B). Is that an artifact from refinement strategy/or coloring scale? I think authors should discuss something related to this considerable difference.

The average B-factors for the two data sets are 20 (F2C) and 35 (F2N) (angstrom²). We hypothesize that the higher B-factors for F2N could be due to the poor quality of the diffraction for F2N after a 90 degree rotation in phi, leading to the higher temperature factor. We provide visualization of the diffraction patterns below. We have added this interpretation to the supplementary text.

F2N – 1.58Å structure

Poor quality during 90° rotation

F2C – 1.46Å structure

No change in diffraction quality after 90° rotation

- Could authors add a label that indicates the up panels on Fig. S19A are the top view of the barrels, and below is the bottom one? It would be helpful for readers.

Updated, as suggested for Figures S18A and S19A.

- Crystallographic parameters for F2N are missing (table S5).

F2N is now deposited and crystallographic parameters have been added to old Table S5 (new Table S9).

- All designed sequences for TIM barrels and other topologies have to be reported (even the non-well behaved designs). In table S4 there are only some sequences reported.

All sequences are given in the Supplementary Dataset S2 and S3, including all designs generated by the model, as well as the subset of designs experimentally tested. We have added additional Supplementary Tables S2-S5 showing the sequence alignment for native test case designs.

Some minor issues:

- What do the authors mean by learned degeneracy? Do they refer to the ability of elements structurally different but with the ability to yield the same output?

This is written in the context of the classifier performance. We show that the classifier often confuses biochemically similar residue types. We also know that proteins are tolerant to mutations, in particular ones between similar residues, such as mutating a tyrosine to a phenylalanine. So, although during training there is a single ground-truth label for each residue position, in effect the classifier learns to predict higher entropy distributions. We have reworded this slightly to be more clear.

- The baseline model accuracy of 33.5% is a standard/normal value? Maybe it could be briefly commented on for non-expert readers.

The baseline model is the model that predicts residues/rotamers from backbone atoms only.

- Even the approach is promising, how can the authors anticipate it could be used to design water-mediated contacts, interfaces, etc.? Or did they mean using a different training set?

Certain elements of the approach can transfer to the task of designing residues with water-mediated contacts. However, we don't elaborate on this in the paper, so we have omitted this statement for now.

- Reference for NovoTIMs should be updated to the current publication (JMB, 2021).

Updated, as suggested.

Also, names changed from NovoTIMs to DeNovoTIMs and can be updated in this manuscript to consistency.

Updated, as suggested.

- Why is reported NovoTIM14a and b?

This is because NovoTIM14 is two-fold symmetric not four-fold, so we compare the subunits of the four-fold symmetric designs with each of the two subunits for NovoTIM14. We have updated the figure legends to include this information.

- Protein purification, italics in *E. coli*.

Corrected.

- In methods, CD melting curves interval is from 25-90 °C even the plots are reported until 95 °C.

Corrected.

- In the crystallography methods, criteria for resolution cutoff have to be mentioned.

We have added this information to the Methods section. We used I/σ minimum 1.5 and CC(1/2) above 70%

Also, the PDB ID for F2N is missing; it's just indicated XXX.

At the time of submission, we had not yet deposited F2N. We have done so now and its ID is 7SMJ.

- In the crystallographic table (table S5), some superscripts are missing on the table footnote. In addition, resolution limits for the outer shell should be reported.

Updated, as suggested.

- Why in figure S11, left panel for 1acf, the line indicates elution volume around 14 mL and not more to the right that matches with the elution of native protein? The same applies to the right panel (3mx7).

The line indicates the expected elution volume based on the native protein molecular weight (MW), as we describe in the figure legend. Both 1acf and 3mx7 elute late off of SEC relative to the expected volume. 3mx7 has hydrophobic surface residues that likely interact with the column, possibly leading it to elute later.

Reviewers' Comments:

Reviewer #1:

Remarks to the Author:

Thank you for all the corrections and clarifications. The paper should now be in good shape for publishing.

Reviewer #2:

Remarks to the Author:

The authors have addressed my questions well and made changes where it is applicable, and I totally understand that the authors have tried to solve structures for all designed proteins but some of them may not be crystalized. I recommend the manuscript to be accepted.

Reviewer #3:

Remarks to the Author:

The revised manuscript by Anand-Achim and colleagues has been substantially improved and shows a clear and compelling story. All of my previous comments and suggestions were successfully addressed. I appreciate their replies and the extra details included in the new version (methodological and discussion). Therefore, I support the publication of the manuscript in the current version.